# Pin the Tail on the Model: Blindfolded Repair of User-Flagged Failures in Text-to-Image Services

**Gefei Tan**[*]
Northwestern University

**Ali Shahin Shamsabadi**[*]
Brave Software

**Ellen Kolesnikova**[†]
Decatur High School

**Hamed Haddadi**
Brave Software & Imperial College London

**Xiao Wang**
Northwestern University

## Abstract

Diffusion models are increasingly deployed in real-world text-to-image services. These models, however, encode implicit assumptions about the world based on web-scraped image-caption pairs used during training. Over time, such assumptions may become outdated, incorrect, or socially biased–leading to failures where the generated images misalign with users' expectations or evolving societal norms. Identifying and fixing such failures is challenging and, thus, a valuable asset for service providers, as failures often emerge post-deployment and demand specialized expertise and resources to resolve them. In this work, we introduce *SURE*, the first end-to-end framework that **SecU**rely **RE**pairs failures flagged by users of diffusion-based services. *SURE* enables the service provider to securely collaborate with an external third-party specialized in model repairing (i.e., Model Repair Institute) without compromising the confidentiality of user feedback, the service provider's proprietary model, or the Model Repair Institute's proprietary repairing knowledge. To achieve the best possible efficiency, we propose a co-design of a model editing algorithm with a customized two-party cryptographic protocol. Our experiments show that *SURE* is highly practical: *SURE* securely and effectively repairs all 32 layers of Stable Diffusion v1.4 in under 17 seconds (four orders of magnitude more efficient than a general baseline). Our results demonstrate that practical, secure model repair is attainable for large-scale, modern diffusion services.

## 1 Introduction

A growing number of real-world services [26, 29, 1, 25, 40, 2, 6, 21, 24, 28, 17] are helping millions of users to create images from textual prompts [45]. These services are typically powered by test-to-image diffusion models [19, 39], which generate high-quality images [45, 7] when trained on billion-scale datasets of image-caption pairs scraped from the web. However, diffusion models implicitly encode the knowledge and assumptions present in their training data [15, 31, 3, 38, 8], which then appear again during image generation. This can lead to unintentional failures: although the generated image may be high quality and technically accurate, it can still misalign with users' values and expectations. For example, diffusion models might retain outdated or incorrect information (e.g., the identity of a country's president or a celebrity's hairstyle). More importantly, diffusion models may encode harmful stereotypical assumptions about professions into their parameters. For example, when given the prompt "A photo of a CEO", the commercial image generation services predominantly generate images of men—only 4% of outputs depict women [30].

---

[*]Contributed equally as co-first authors.
[†]Work done during an internship at Northwestern University.

39th Conference on Neural Information Processing Systems (NeurIPS 2025).

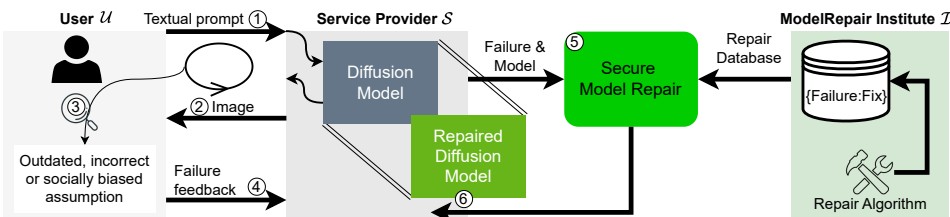

Figure 1: **Block diagram of *SURE*.** A service provider $\mathcal{S}$ deploys a diffusion model to generate images (services) in response to textual prompts (user queries) (① & ②). When a user $\mathcal{U}$ notices a failure in $\mathcal{S}$'s services–due to the outdated, incorrect or discriminative assumption– (③), $\mathcal{U}$ provides feedback to $\mathcal{S}$ (④). $\mathcal{S}$ then collaborates with a ModelRepair Institute to securely repair the model (⑤) through cryptographic protocols that preserve the confidentiality of users' feedback, service provider's model and institute's proprietary repairing knowledge. Finally, the repaired model is returned only to the service provider (⑥).

When model failure happens in practice, users typically discover these failures and provide feedback on the current behavior of models to the service providers [10]. However, it is challenging for service providers to incorporate feedback and repair their models for several reasons. First, these unintentional failures emerge over time [31] as world knowledge or societal norms evolve. Second, repairing diffusion models usually requires substantial expertise and resources [9, 4], which service providers, especially start-ups, lack. One possible solution to address this problem is for the service provider to share its model and user feedback with an external institution specializing in model repair[3] who can repair the failure. However, this approach raises significant concerns for all parties involved. Sharing the model compromises its confidentiality, undermining the commercial value of service provider's image-generation service. Sharing user feedback is also not permissible due to privacy regulations such as the GDPR [41]. Meanwhile, the repair institution is reluctant to disclose its repair techniques in order to protect its own intellectual property.

**Our Work.** We address these challenges by introducing *two-party secure repairing based on user feedback*. We propose *SURE* (Figure 1), a secure framework that enables a service provider and an external model repair institution to collaboratively repair the service provider's diffusion model using users' feedback and the institution's repair expertise while remaining mutually blindfolded. To ensure that the users' feedback, the provider's proprietary model, and the expert's repair recipe all remain confidential, *SURE* leverages secure two-party computation (2PC) techniques [46, 16], which allow two parties to jointly compute any function without revealing anything about their private data beyond the function output. Directly computing an existing knowledge editing algorithm [30, 3, 15, 45] in 2PC is theoretically feasible, but becomes completely unrealistic in practice due to the high computation cost of both 2PC and knowledge editing algorithms. Instead, we take a co-design approach to jointly optimize both the machine learning and cryptography components. *SURE* targets and updates only a tiny fraction of parameters–namely, the keys and values of cross-attention layers– with a crypto-friendly repair formula. Our design enables each party to shift expensive operations offline, allowing us to design a lightweight, customized cryptographic protocol on top of it.

Our protocol consists of (1) a small 2PC circuit that privately matches the user feedback to the most relevant fix and (2) an oblivious-transfer-based protocol [32] that securely delivers the corresponding fix. Our protocol completely avoids matrix operations inside 2PC, and the cryptographic overhead remains constant regardless of the number of layers repaired. Our end-to-end secure repair framework is highly efficient and scalable: our experiments show that a service provider can use *SURE* to repair all 32 layers of their Stable Diffusion v1.4 [34] in collaboration with a Model Repair Institute in under 17 seconds, whereas an optimized baseline protocol needs over 100 hours.

In summary, we propose the first secure repairing framework that enables users to control the model's behavior over time and enables service providers to ensure continued alignment with social expectations post-training without retraining. We highlight the following contributions:

- We **initiate the study of an important and emerging problem of model repair while protecting the security of the model, data, and the repairing knowledge**. We formulate the security and utility requirements needed in real-world applications.

---

[3] https://humanfeedback.io/

- Although generic cryptographic protocols can be used to support this task, their efficiency is completely unacceptable for realistic applications. To this end, **we co-design a crypto-friendly editing algorithm and a customized 2PC protocol** such that the editing algorithm is as effective as state-of-the-art model repair approaches while minimizing the protocol cost when executed using our optimized cryptographic protocol.

- We implemented our protocol and a baseline protocol using generic 2PC. We tested their performance for repairing Stable Diffusion v1.4. We observed **4 orders of magnitude improvement in runtime** compared to the baseline, bringing secure model repairing from merely a concept to something that can practically be deployed on modern models.

## 2   Notations and Preliminaries

**Notations.** We use lowercase bold letters like $\mathbf{c}$ to denote column vectors and uppercase bold letters like $\mathbf{W}$ to denote matrices. We write $[n]$ to denote the set $\{1, 2, \ldots, n\}$. We use consistent notation for values in the diffusion model architecture, as defined in the next few paragraphs.

**Diffusion Models** [19, 39] are a class of generative models that have recently emerged as the SOTA in image generation. Inspired by non-equilibrium thermodynamics, diffusion models use a fixed algorithm to incrementally add random noise to images (or other data), and then learn how to reverse this process. The learned model is then used for image generation. Diffusion models have not always been the SOTA in image generation; prior to diffusion models, GANs were the most promising image generation models [11]. However, compared to GANs, diffusion models offer multiple advantages that lead to better results [12]. Diffusion models use more stable loss metrics than GANs. Additionally, because diffusion models generate images over a series of timesteps, their task is easier than that of GANs, which do it in one pass.

In this work, we focus on **text-to-image diffusion models** [33, 36, 27, 18, 35], where the diffusion process is guided by a user-provided text prompt that is embedded and injected into the cross-attention layers of the model. Formally, we consider a diffusion model $\mathcal{M}$ that generates images by denoising a Gaussian sample $\mathbf{x}_T$ over $T$ time steps using a neural network $D_\theta(\mathbf{x}_t, t, c)$, where $c$ is a conditioning signal derived from the text. The text prompt is first tokenized and processed by a text encoder, which outputs a sequence of token embeddings $\{\mathbf{c}_i\}_{i=1}^{\ell}$ where $\mathbf{c}_i \in \mathbb{R}^c$ that represent the semantic content of the input text. Let $\mathbf{C} = [\mathbf{c}_1, \ldots, \mathbf{c}_\ell] \in \mathbb{R}^{c \times \ell}$ denote the resulting matrix of text embeddings. At every cross-attention layer, these embeddings are linearly projected into $\mathbf{K} = \mathbf{W}_K \mathbf{C} \in \mathbb{R}^{\ell \times k}$ and $\mathbf{V} = \mathbf{W}_V \mathbf{C} \in \mathbb{R}^{\ell \times v}$ using learned *key* and *value* projection matrices $\mathbf{W}_K \in \mathbb{R}^{k \times c}$ and $\mathbf{W}_V \in \mathbb{R}^{v \times c}$, respectively. Next, the key $\mathbf{K}$ is multiplied by a query $\mathbf{Q} \in \mathbb{R}^{n \times k}$ that represents the current image's visual feature. The cross-attention mechanism computes an attention map and a weighted value output: $\mathbf{M} = \text{softmax}\left(\frac{\mathbf{Q}\mathbf{K}^\top}{\sqrt{m}}\right)$ and $\mathbf{O} = \mathbf{M}\mathbf{V}$. The output $\mathbf{O}$ guides the visual features based on the semantic content of the text prompt.

**Diffusion Model Editing** aims to remove various biases from diffusion models and has become increasingly important as these models gain widespread adoption. One way it is done is by adjusting various aspects of the training process to limit bias; this can include altering the loss function [37] or debiasing the training dataset [23]. Fine-tuning existing diffusion models is perhaps a more realistic approach, as biases can become apparent after training. To do this, a small fraction of the weights in the diffusion model are updated to fix a specific problem. This can be done by editing the text encoder [44], or by directly editing the diffusion model [13, 30]. We focus on fine-tuning after training in this paper, as this ensures models can be updated as needed and do not need to be retrained.

**Oblivious Transfer** (OT) is a fundamental cryptographic primitive essential for secure computation protocols [32]. In a 1-out-of-$n$ OT, a sender possesses $n$ messages $(m_1, \ldots, m_n)$, and a receiver selects an index $i \in [n]$ to retrieve $m_i$ without revealing $i$ to the sender. Simultaneously, the receiver gains no information about the other messages $m_j$ for $j \neq i$. This ensures that the sender remains oblivious to the receiver's choice, and the receiver learns only the selected message.

**Secure Two-Party Computation** (2PC) [46, 16] enables two mutually distrustful parties, each holding private inputs, to jointly compute a public function without revealing any information beyond the output. We consider 2PC in the presence of static semi-honest adversaries, where parties follow the protocol but may attempt to learn additional information from the protocol execution transcript. The ideal functionalities of 1-out-of-$n$ OT and 2PC are presented in Appendix B.

# 3   Problem Description

**Parties and Trust Assumptions.** We consider a setting including three parties as illustrated in Figure 1: a **service provider** $\mathcal{S}$ that offers diffusion-based image generation services, **users** $\mathcal{U}$ who query the service and provides feedback when observing service failures, and a **model repair institute** $\mathcal{I}$, which specializes in repairing model failure and collaborates with $\mathcal{S}$ to repair its model.

In this setting, we make the following trust assumptions in our threat model:

- *Users* $\mathcal{U}$ query the image generation service with their textual prompt and receive images. $\mathcal{U}$ discovers failures as they use $\mathcal{M}$-based services of $\mathcal{S}$. $\mathcal{U}$'s flagged failures are because of the fact that $\mathcal{M}$ acquires knowledge within their training data [30] which become outdated, incorrect and harmful over time. For instance, for the prompt "A photo of a CEO", only 4% of generated images (with random seeds) contain female figures [30]. **This feedback should only be visible to $\mathcal{S}$.**

- *A Service Provider* $\mathcal{S}$ trains the text-to-image diffusion model $\mathcal{M}$ on huge amounts of web-scraped image-caption pairs, and provides image generation services using $\mathcal{M}$. $\mathcal{S}$ **wants to protect (i) the proprietary weights of** $\mathcal{M}$ **and (ii) user-submitted feedback**, which may contain sensitive user data and is subject to privacy regulations. We additionally require $\mathcal{S}$ **must not reveal which failure it is fixing when interacting with** $\mathcal{I}$, as it might inadvertently leak user data.

- *A Model Repair Institute* $\mathcal{I}$ specialized in repairing text-to-image diffusion models. $\mathcal{I}$ **wants to keep both its repairing algorithm and fix database secret**, as they are its core intellectual property.

**Goal and Technical Challenges**. The above-mentioned failures make the world knowledge of $\mathcal{M}$ in deployments unaligned with users' values and expectations [15, 31, 3, 38, 8]. Our goal is to repair $\mathcal{M}$ failures identified by $\mathcal{U}$. Although users are essential for flagging failures, they do not directly participate in the repair process. Once the feedback is submitted, it becomes the responsibility of $\mathcal{S}$ to repair their model. As service providers usually lack expertise and resources for repairing failures (they mostly focus on enhancing image qualities), $\mathcal{S}$ needs to contact an external Model Repair Institute $\mathcal{I}$ to perform such fixes. This is challenging for several reasons. First, service providers are not allowed to share user's data[4] with third-parties due to privacy regulations. Second, service providers are not willing to hand over their models to third parties due to IP concerns. Third, Model Repair Institutes are not willing to disclose their fixes to service providers to protect their business model. Therefore, we model the protocol as a two-party computation between $\mathcal{S}$ and $\mathcal{I}$, with the feedback treated as private input held by $\mathcal{S}$. We adopt the standard *semi-honest security* model (see Appendix D for the extension to malicious security), where both parties follow the repair protocol correctly but may try to infer additional information from the interaction: both $\mathcal{S}$ and $\mathcal{I}$ are institutions with legal and reputational reasons to behave correctly during model repairing, though they may have incentives to recover more information.

**Our Solution.** Given the trust assumptions above, our goal is to build a provably secure protocol that protects the private inputs of both parties: the model weights and user feedback held by $\mathcal{S}$, and the proprietary repair logic and database held by $\mathcal{I}$. To achieve this, we rely on cryptographic techniques. We design a crypto-friendly knowledge editing algorithm by adapting an efficient editing method that avoids retraining from scratch. Based on this, we construct a lightweight, customized two-party computation protocol, which we detail in the next section.

# 4   *SURE*: SecUre model REpairing

We propose *SURE*, a protocol for effective, efficient, and secure repair of text-to-image diffusion models based on user feedback and collaboration between a service provider $\mathcal{S}$ and a model repair institute $\mathcal{I}$. *SURE* combines a crypto-friendly model repair algorithm with a customized two-party computation (2PC) protocol. Our approach builds on recent knowledge editing techniques [30] that enable model updates without full retraining. However, applying these techniques out-of-the-box is unsuitable for efficient 2PC due to the large number of layers in diffusion models and the high cost of interactive operations such as high-dimensional matrix multiplications and inverses. Our key insight is that most of this cost can be avoided by carefully modifying the editing algorithm.

---

[4]Note that we do not consider protection of confidentiality or privacy of users' request to $\mathcal{S}$ as it is only a failure identification and their institution knows their data, but we want to protect it against other institutions.

---

**Algorithm 1:** Repair database creation

---

**Input:** A Model Repair Institute $\mathcal{I}$, A public text encoder (*TextEncoder*), A collection of Failures
**Output:** Repair Database

1: Repair Database $= \{\}$
2: **for all** failure $\in$ Failures **do**
3:    Repair data pair $= \{\text{source prompt:} \mathbf{x}, \text{destination prompt:} \mathbf{x}'\}$    ▷ `Creating a repairing data`
4:    $\{\mathbf{C} \in \mathbb{R}^{c \times l}, \mathbf{C}' \in \mathbb{R}^{c \times l'}\} = \textit{TextEncoder}(\{\mathbf{x}, \mathbf{x}'\})$    ▷ `Tokenizing and computing embeddings`
5:    $\mathbf{C}^* \in \mathbb{R}^{c \times l} = \textit{RemoveAdditionalTokens}(\mathbf{C}')$ ▷ `Creating an embedding that corresponds to`
    `the same source token by discarding the embedding of additional tokens in the`
    `destination prompt`
6:    $\mathbf{W}_{\text{fix}} = \left(\lambda_{\text{failure}}\mathbf{I} + \mathbf{C}^*\mathbf{C}^\top\right)\left(\lambda_{\text{failure}}\mathbf{I} + \mathbf{C}\mathbf{C}^\top\right)^{-1}$    ▷ `Creating Repair Knowledge`
7:    Repair Database.append($\{\text{failure} : \mathbf{W}_{\text{fix}}\}$)
8: Output Repair Database

---

---

**Algorithm 2:** Repair diffusion model parameters

---

**Input:** Service Provider $\mathcal{S}$, A text-to-image diffusion model $\mathcal{M}$, Received repair knowledge $\mathbf{W}_{\text{fix}}$
**Output:** Updated parameters of the repaired text-to-image diffusion model $\mathcal{M}$

1: CrossAttentionLayers $\leftarrow$ *CrossAttentionAccess*($\mathcal{M}$)    ▷ `Extract cross-attention layers that`
    `map textual data into visual data`
2: **for all** i $\in$ Size(CrossAttentionLayers) **do**
3:    $\mathbf{W}_V'^i \leftarrow \mathbf{W}_V^i \mathbf{W}_{\text{fix}}$    ▷ `Update value projection matrix`
4:    $\mathbf{W}_K'^i \leftarrow \mathbf{W}_K^i \mathbf{W}_{\text{fix}}$    ▷ `Update key projection matrix`
5: Updated diffusion model returned to only the service provider

---

We design a crypto-friendly repair algorithm (Section 4.1) tailored to the 2PC setting, without compromising the effectiveness of the original editing method. Our redesigned algorithm shifts almost all heavy computation offline, allowing each party to process its data locally and independently. Specifically: (1) $\mathcal{I}$ constructs the repair database offline (Algorithm 1); and (2) $\mathcal{S}$ applies the fix to its model parameters locally (Algorithm 2). In the online phase, we further develop a custom 2PC protocol (Section 4.2) that enables the service provider to securely locate and receive the fix corresponding to their failures from the institute's repair database through a secure fuzzy matching procedure and a lightweight Oblivious Transfer (OT) protocol. We prove (Section 4.3) that our protocol keeps users' feedback, service provider's model parameters, and the institute's proprietary editing algorithm confidential while ensuring that the model is faithfully repaired.

## 4.1 Crypto-Friendly Model Repair Algorithm

We instantiate *SURE* based on the *Text-to-Image Model Editing* (TIME) procedure introduced by Orgad *et al.* [30]. We briefly review their core editing algorithm before presenting our modifications.

The editing algorithm in TIME takes as input two prompts:

- A *source prompt*, e.g., "a photo of CEO" that under-specifies certain visual attributes. It allows the model to fill in missing details using its implicit assumptions, which could reflect bias.
- A more specific *destination prompt*, e.g., "a photo of **female** CEO" where an explicit attribute is added to correct the failure in the original source prompt.

The editing goal is to repair failures in the model's original output by shifting the image generation from reflecting the source prompt to better align with the intended visual attributes of the destination prompt. This enables targeted correction of outdated, incorrect, or socially biased associations embedded in the model. The key insight from Orgad *et al.* is that it suffices to update only the key and value projection matrices $\mathbf{W}_K$ and $\mathbf{W}_V$ (see Section 2 for detailed definitions) within the model's cross-attention layers. These matrices are responsible for mapping textual tokens into attention-compatible visual representations, and patching them effectively alters the generated output.

Let $\{\mathbf{c}_i\}_{i=1}^{\ell} \subset \mathbb{R}^c$ and $\{\mathbf{c}_j'\}_{j=1}^{\ell'} \subset \mathbb{R}^c$ be the token embeddings of the source and destination prompt. For every source token, TIME locates the corresponding destination token that contains the same word and denotes its embedding by $\mathbf{c}_i^*$. This gives the aligned set $\{\mathbf{c}_i^*\}_{i=1}^{\ell}$ for tokens appear in both prompts. Let $\mathbf{C} = [\mathbf{c}_1, \ldots, \mathbf{c}_{\ell}]$ and $\mathbf{C}^* = [\mathbf{c}_1^*, \ldots, \mathbf{c}_{\ell}^*]$, for every layer $i$, the closed-form update

formula (Equation 5 in [30]) is given by

$$\mathbf{W}_K^{\prime i} = \left(\lambda\,\mathbf{W}_K^i + \mathbf{K}^*\mathbf{C}^\top\right)\left(\lambda\,\mathbf{I}_d + \mathbf{C}\mathbf{C}^\top\right)^{-1} \& \mathbf{W}_V^{\prime i} = \left(\lambda\,\mathbf{W}_V^i + \mathbf{V}^*\mathbf{C}^\top\right)\left(\lambda\,\mathbf{I}_d + \mathbf{C}\mathbf{C}^\top\right)^{-1}, \quad (1)$$

where $\lambda \in \mathbb{R}^+$ is a hyperparameter, and $\mathbf{K}^* = \mathbf{W}_K\mathbf{C}^*$ and $\mathbf{V}^* = \mathbf{W}_V\mathbf{C}^*$.

**Efficiency and Privacy Challenges.** The most direct way to securely evaluate the above update formula is to encode it as a circuit and run a generic 2PC: the provider $\mathcal{S}$ supplies the private weights $\mathbf{W}_\star^i$, the institute $\mathcal{I}$ supplies $\mathbf{C}, \mathbf{C}^*, \lambda$, and the circuit outputs the updated weights $\mathbf{W}_\star^{\prime i}$ to $\mathcal{S}$. Despite significant advances in modern 2PC protocols [43, 20], applying them directly to this task remains inefficient. To illustrate the limitations of this generic approach, we implemented a baseline that computes the editing algorithm in a generic 2PC protocol, and it requires **over 100 hours** to perform a single repair (see Section 5). More importantly, because generic 2PC assumes the circuit is public, service provider will always learn the institute's proprietary repair algorithm.

**Our Crypto-Friendly Editing Formula.** The bottleneck above mainly comes from forcing heavy matrix operations into the secure computation. Our key observation is that we can refactor the editing formula in a way that completely eliminates any matrix operations inside 2PC. The matrix update formula in Equation 1 can be refactored as follow:

$$
\begin{aligned}
\mathbf{W}_V^{\prime i} &= \left(\lambda\,\mathbf{W}_V^i + \mathbf{V}^*\mathbf{C}^\top\right)\left(\lambda\,\mathbf{I}_d + \mathbf{C}\mathbf{C}^\top\right)^{-1} \\
&= \left(\lambda\,\mathbf{W}_V^i + \mathbf{W}_V^i\mathbf{C}^*\mathbf{C}^\top\right)\left(\lambda\,\mathbf{I} + \mathbf{C}\mathbf{C}^\top\right)^{-1} \\
&= \underbrace{\mathbf{W}_V^i}_{\text{known to }\mathcal{S}} \underbrace{\left(\lambda\,\mathbf{I} + \mathbf{C}^*\mathbf{C}^\top\right)\left(\lambda\,\mathbf{I} + \mathbf{C}\mathbf{C}^\top\right)^{-1}}_{\mathbf{W}_{\text{fix}}, \text{ known to }\mathcal{I}}.
\end{aligned} \quad (2)
$$

Here, $\mathcal{S}$ holds $\mathbf{W}_V$, and $\mathcal{I}$ holds $\mathbf{C}, \mathbf{C}^*$, and the hyperparameter $\lambda$. Thus, $\mathcal{I}$ can compute

$$\mathbf{W}_{\text{fix}} \leftarrow \left(\lambda\,\mathbf{I} + \mathbf{C}^*\mathbf{C}^\top\right)\left(\lambda\,\mathbf{I} + \mathbf{C}\mathbf{C}^\top\right)^{-1}, \quad (3)$$

and $\mathcal{S}$ can update the matrix by computing $\mathbf{W}_V^{\prime i} \leftarrow \mathbf{W}_V^i\mathbf{W}_{\text{fix}}$. The above refactored equation applies identically to $\mathbf{W}_K$ and holds for all layers in the model. The fix matrix $\mathbf{W}_{\text{fix}}$ now encapsulates the semantics of the update and fully decouples model-specific parameters from repairs. Our refactored formula yields three immediate advantages for our purposes:

- **One fix fits all.** The same $\mathbf{W}_{\text{fix}}$ matrix can be reused across every cross-attention layer $i$, and applies uniformly to both $\mathbf{W}_K^i$ and $\mathbf{W}_V^i$. This significantly simplifies the repair process and reduces communication.

- **Matrix algebra disappears from 2PC.** All matrix operations to compute $\mathbf{W}_{\text{fix}}$ are handled entirely by $\mathcal{I}$ offline. Then, $\mathcal{S}$ can use lighter cryptographic primitives like OT to acquire the $\mathbf{W}_{\text{fix}}$ from $\mathcal{I}$.

- **Algorithm privacy is preserved.** Because the fix is provided as a single matrix and applied independently by $\mathcal{S}$, there is no need to reveal the full structure of the editing algorithm or encode it into a shared circuit. Therefore, $\mathcal{I}$'s proprietary repair method remains hidden from $\mathcal{S}$.

We adopt this new editing formula in our protocol *SURE*. As we show in Section 5, this seemingly simple refactor achieve four orders of magnitude speed up over the baseline. Appendix C describes the class of editing algorithms that our framework supports without incurring any utility loss.

## 4.2 Efficient Two-Party Model Repair Protocol

We now provide a detailed description of the secure two-party model repair protocol in *SURE*. The ideal functionality and our two-party model repair protocol are presented in Figure 2 and Figure 3.

We first briefly recall our setting. The protocol *SURE* involves two parties: a service provider $\mathcal{S}$ and a model repair institute $\mathcal{I}$. $\mathcal{S}$ wish to repair its deployed model $\mathcal{M}$ and derives from aggregated user feedback a query key $\mathbf{k}_{\text{qry}} \in \mathbb{R}^k$ that captures the failure domain to be fixed. $\mathcal{I}$ maintains a private key–value repair database $\{k_j : \mathbf{W}_{\text{fix},j}\}_{j \in [n]}$ of size $n$, where each key $\mathbf{k}_i \in \mathbb{R}^k$ semantically labels a failure and each $\mathbf{W}_{\text{fix},j}$ is the repair matrix for this failure. The protocol consists of three stages:

1. **Database Initialization.** Before interacting with $\mathcal{S}$, the institute $\mathcal{I}$ locally computes $\mathbf{W}_{\text{fix},j}$ from the embedding matrices $\mathbf{C}_j, \mathbf{C}_j^*$ and edit hyperparameter $\lambda_j$, and tag the fix with a key $\mathbf{k}_j$ that semantically describe the failure.



**Functionality $\mathcal{F}_{\text{Repair}}$**

This functionality is parameterized by a similarity metric $d(\cdot, \cdot)$ and a database size $n$.

**Input:**
- $\mathcal{S}$ inputs a query key $\mathbf{k}_{\text{qry}} \in \mathbb{R}^k$ and model matrices $\{\mathbf{W}_V^i \in \mathbb{R}^{v \times c}, \mathbf{W}_K^i \in \mathbb{R}^{k \times c}\}_{i \in [m]}$.
- $\mathcal{I}$ inputs the database $\{\mathbf{k}_i, \mathbf{C}_i, \mathbf{C}_i^*, \lambda_i\}_{i \in [n]}$ where $\mathbf{C}_i, \mathbf{C}_i^* \in \mathbb{R}^{c \times l}$, $\lambda_i \in \mathbb{R}^+$, and $\mathbf{k}_i \in \mathbb{R}^k$.

**Model Repair:**
1. Compute $p = \arg\min_{i \in [n]} d(\mathbf{k}_{\text{qry}}, \mathbf{k}_i)$, breaking ties by choosing the smallest $i$.
2. For each model layer $i \in [m]$, compute and send the following updated matrices and index $p$ to $\mathcal{S}$:

$$\mathbf{W}_\star^{\prime i} \leftarrow \left(\lambda_p \mathbf{W}_\star^i + \mathbf{W}_\star^i \mathbf{C}_p^* \mathbf{C}_p^\top\right)\left(\lambda_p \mathbf{I}_c + \mathbf{C}_p \mathbf{C}_p^\top\right)^{-1} \quad \star \in \{V, K\}.$$



Figure 2: Ideal functionality of model repair between $\mathcal{S}$ and $\mathcal{I}$.



**Protocol $\Pi_{\text{Repair}}$**

**Input:**
- The service provider $\mathcal{S}$ and institute $\mathcal{I}$ agree on a similarity metric $d(\cdot, \cdot)$ and the database size $n$.
- $\mathcal{S}$ inputs fix query vector $\mathbf{k}_{\text{qry}} \in \mathbb{R}^k$ and model matrices $\{\mathbf{W}_V^i \in \mathbb{R}^{v \times c}, \mathbf{W}_K^i \in \mathbb{R}^{k \times c}\}_{i \in [m]}$, where $m$ is the total number of model layers.
- $\mathcal{I}$ inputs $\{\mathbf{k}_j, \mathbf{C}_j, \mathbf{C}_j^*, \lambda_j\}_{j \in [n]}$, where $\mathbf{C}_j, \mathbf{C}_j^* \in \mathbb{R}^{c \times l}$, $\lambda_j \in \mathbb{R}^+$, $\mathbf{k}_j \in \mathbb{R}^k$, and $n$ is the size of repair database.

**Database Initialization:** $\mathcal{I}$ computes the repair database $\{\mathbf{k}_i : \mathbf{W}_{\text{fix,j}}\}_{j \in [n]}$, where

$$\mathbf{W}_{\text{fix,j}} \leftarrow \left(\lambda_j \mathbf{I}_c + \mathbf{C}_j^* \mathbf{C}_j^\top\right)\left(\lambda_j \mathbf{I}_c + \mathbf{C}_j \mathbf{C}_j^\top\right)^{-1}.$$

**Matching:** Let $\mathcal{C}_{d,n}$ be the circuit that outputs$(p, \perp)$, where $p = \arg\min_{j \in [n]} d(\mathbf{k}_{\text{qry}}, \mathbf{k}_j)$, breaking ties by choosing the smallest $j$. $\mathcal{S}$ and $\mathcal{I}$ send $(\mathcal{C}_{d,n}, \mathbf{k}_{\text{qry}})$ and $(\mathcal{C}_{d,n}, (\mathbf{k}_1, \ldots, \mathbf{k}_n))$ to $\mathcal{F}_{\text{2PC}}$. $\mathcal{S}$ receives the fix matrix index $p$.

**Model Repair:**
1. $\mathcal{S}$ and $\mathcal{I}$ send $(\text{recv}, n, p)$ and $(\text{send}, n, \{\mathbf{W}_{\text{fix,j}}\}_{j \in [n]})$ to $\mathcal{F}_{\text{OT}}$. $\mathcal{S}$ obtains the fix matrix $\mathbf{W}_{\text{fix,p}}$.
2. For each model layer $i \in [m]$ and $\star \in \{K, V\}$, $\mathcal{S}$ locally updates each layer of its model using the same fix matrix: $\mathbf{W}_\star^{\prime i} \leftarrow \mathbf{W}_\star^i \mathbf{W}_{\text{fix,p}}$.



Figure 3: Our secure model repair protocol in the $(\mathcal{F}_{\text{OT}}, \mathcal{F}_{\text{2PC}})$-hybrid model.

2. **Matching.** $\mathcal{S}$ and $\mathcal{I}$ run a small circuit inside 2PC to locate the database entry whose key $\mathbf{k}_p$ minimizes a public similarity metric $d(\mathbf{k}_{\text{qry}}, \mathbf{k}_j)$. After this stage, only $\mathcal{S}$ learns the index $p$; $\mathcal{I}$ learns nothing about $\mathbf{k}_{\text{qry}}$ beyond the fact that a comparison occurred. When an exact match is sufficient—e.g., $d$ is the discrete metric or the database indexes are public, $\mathcal{I}$ can determine $p$ outright, so this stage can be skipped and the parties proceed directly to the next step.

3. **Oblivious Model Repair.** After acquiring the index $p$, $\mathcal{I}$ runs an OT protocol to retrieve the single matrix $\mathbf{W}_{\text{fix,p}}$ without revealing $p$ and without accessing any other entry. It then updates every cross-attention layer locally by right-multiplying both value and key projections with $\mathbf{W}_{\text{fix,p}}$ to complete the repair.

**Security Guarantees.** Our protocol ensures that (i) the institute $\mathcal{I}$ learns nothing about the model $\mathcal{M}$ or the query key $\mathbf{k}_{\text{qry}}$; (ii) the service provider $\mathcal{S}$ learns only the single fix matrix matching its query and gains no information about any other entry in $\mathcal{I}$'s database; and (iii) the editing algorithm itself remains private, because $\mathcal{I}$ builds the database offline, the editing algorithm chosen by $\mathcal{I}$ remains entirely hidden from $\mathcal{S}$. In the next section, we formalize and prove these guarantees.

### 4.3 Security Proof

In this section, we establish the security of our protocol $\Pi_{\text{Repair}}$ (Figure 3) and show how it can be generalized to any editing mechanism while hiding the editing algorithm being employed by the institute. All of our proofs are based on the standard composition paradigm [5]. We now state the following main security theorem of our protocol.

**Theorem 1** (Protocol Security). *Protocol $\Pi_{\text{Repair}}$ (Figure 3) securely realizes $\mathcal{F}_{\text{Repair}}$ (Figure 2) in the $(\mathcal{F}_{\text{OT}}, \mathcal{F}_{\text{2PC}})$-hybrid model against semi-honest adversaries.*

*Proof.* For clarity, we denote the service provider by $P_1$ and the repair institute by $P_2$ for the remainder of the proof.

**Correctness.** Note that all matrix products in both the protocol and the functionality are well-defined. Additionally, for all $j \in [n]$, the regularization parameter $\lambda_j > 0$, hence the matrix $(\lambda_j \mathbf{I}_c + \mathbf{C}_j \mathbf{C}_j^\top) \succ 0$ and is therefore invertible.

To prove privacy, we separately consider the case of a corrupted institute and service provider.

**Corrupted Institute $\hat{P}_2$.** It is straightforward to prove security against $\hat{P}_2$, as it receives no output from either $\mathcal{F}_{\mathsf{OT}}$ and $\mathcal{F}_{\mathsf{2PC}}$. Therefore, a simulator $\mathcal{S}_2$ that simply forwards $\hat{P}_2$'s message to $\mathcal{F}_{\mathsf{Repair}}$ can perfectly simulate its view.

**Corrupted Service Provider $\hat{P}_1$.** We construct a simulator $\mathcal{S}_1$ that calls $\hat{P}_1$ as a subroutine and interacts with $\mathcal{F}_{\mathsf{Repair}}$ to simulate its view. $\mathcal{S}_1$ proceeds as follows:

1. $\mathcal{S}_1$ obtains the message $(\mathcal{C}_{d,n}, \mathbf{k}_{\mathsf{qry}})$ from $\hat{P}_1$ and record $\mathbf{k}_{\mathsf{qry}}$.
2. $\mathcal{S}_1$ sends $(\mathbf{k}_{\mathsf{qry}}, \{\mathbf{I}_c, \mathbf{I}_c\})$ to $\mathcal{F}_{\mathsf{Repair}}$ and receives index $p$ and $\{\hat{\mathbf{W}}_V'^i, \hat{\mathbf{W}}_K'^i\}_{i \in [m]}$.
3. $\mathcal{S}_1$ acts as $\mathcal{F}_{\mathsf{2PC}}$ and send $p$ to $\hat{P}_1$; upon obtaining $(\mathsf{recv}, n, p)$ from $\hat{P}_1$, send $\hat{\mathbf{W}}_V'^1$ to $\hat{P}_1$.

We show that $\hat{P}_1$'s view is perfectly simulated. To see this, notice that the ideal world, because $\mathcal{S}_1$ sends identity matrices to $\mathcal{F}_{\mathsf{Repair}}$, for every layer

$$\hat{\mathbf{W}}_V'^i = (\lambda_p \mathbf{I}_c + \mathbf{I}_c \mathbf{C}_p^* \mathbf{C}_p^\top)(\lambda_p \mathbf{I}_c + \mathbf{C}_p \mathbf{C}_p^\top)^{-1} = \mathbf{W}_{\mathsf{fix,p}}.$$

As a result, the matrix $\hat{P}_1$ received in the ideal execution is *exactly* the same from $\mathcal{F}_{\mathsf{OT}}$ in the real execution. Therefore, its view is perfectly simulated. As an honest $P_2$ receives no output in both worlds, the joint output distributions are also identical in both worlds. This concludes the proof. $\square$

**Algorithm Privacy.** For concreteness, we instantiate our protocol based on the editing algorithm of [30]. However, our cryptographic construction readily accommodates *any* repair mechanism: any repair procedure that modifies model weights while leaving the network architecture unchanged can be dropped in without altering the protocol. Moreover, the protocol keeps the institute's choice of editing algorithm confidential. To see this, notice that $\mathcal{S}$ only sees the resulting fix matrix $\mathbf{W}_{\mathsf{fix}}$ while the algorithm itself remains hidden. To formalize this property, we first define the notion of editing algorithms and prove a theorem stating the algorithm-hiding property of our protocol.

**Definition 1** (Editing Algorithms). A model repair editing algorithm is an efficient mapping

$$f : (\mathbf{C}, \mathbf{C}^*, \mathsf{aux}) \to \mathbf{W}_{\mathsf{fix}},$$

where $\mathbf{C}, \mathbf{C}^*$ are the source and target prompt embedding matrices, $\mathsf{aux}$ is institute-held auxiliary input, and $\mathbf{W}_{\mathsf{fix}}$ is the fix matrix of proper dimensions that is right-multiplied to every model layer.

**Theorem 2** (Algorithm Privacy). *Let $U_f = \{f_1, \ldots, f_Z\}$ be any finite family of editing algorithms. Let $\Pi'_{\mathsf{Repair}}$ be the extension of $\Pi_{\mathsf{Repair}}$ in which $\mathcal{I}$ chooses an index $z \in [Z]$, and builds its database using $f_i$. Let $\mathcal{F}'_{\mathsf{Repair}}$ be the corresponding ideal functionality that receives the description of $f_z$ from $\mathcal{I}$, evaluates $f_z$ internally to obtain the fix matrices, and sends those matrices to $\mathcal{S}$. Then, for every PPT adversary $\mathcal{A}_S$ corrupting the service provider $\mathcal{S}$, there exists a PPT simulator $\mathcal{S}'_1$ such that $\mathsf{view}_{\mathcal{A}_S}^{\mathsf{Real}}(\Pi'_{\mathsf{Repair}}) \equiv \mathsf{view}_{\mathcal{S}'_1}^{\mathsf{Ideal}}(\mathcal{F}'_{\mathsf{Repair}})$.*

*Proof.* The proof follows from the security proof of Theorem 1 in a straightforward manner. We define a modified simulator $\mathcal{S}'_1$ that behaves the same as $\mathcal{S}_1$ in the security proof, except it forward $\mathcal{S}$'s message to $\mathcal{F}'_{\mathsf{Repair}}$ instead of $\mathcal{F}_{\mathsf{Repair}}$. It then follows from the security proof that $\Pi'_{\mathsf{Repair}}$ securely realizes $\mathcal{F}'_{\mathsf{Repair}}$ in the $(\mathcal{F}_{\mathsf{OT}}, \mathcal{F}_{\mathsf{2PC}})$-hybrid model and $S'_1$ perfectly simulates $\mathcal{A}_{\mathcal{S}}$'s view. $\square$

Theorem 2 implies that the service provider $\mathcal{S}$ *learns no information* about the institute's chosen editing algorithm $f_i$ beyond what is already implied by the fix matrix $\mathbf{W}_{\mathsf{fix,p}}$. Consequently, our protocol enables the institute to swap in or fine-tune its proprietary repair procedures without touching the underlying cryptographic protocol. This design not only supports fast iteration but also hides the institute's editing knowledge from the service provider.

By contrast, generic secure 2PC protocols would require compiling the entire editing algorithm into a single Boolean or arithmetic circuit that is known to both parties—a standard assumption in secure

**Functionality $\mathcal{F}_{\mathsf{OT}}$**

Upon receiving $(\mathsf{send}, n, \{m_i\}_{i \in [n]})$ from $P_1$ and $(\mathsf{recv}, n, b)$ from the $P_2$ where $b \in [n]$, send $m_b$ to $P_2$.

Figure 4: Ideal functionality of 1-out-of-$n$ oblivious transfer.

**Functionality $\mathcal{F}_{\mathsf{2PC}}$**

For $i \in \{1, 2\}$, upon receiving $(\mathcal{C}, x_i)$ from $P_i$, compute $y_1, y_2 \leftarrow \mathcal{C}(x_1, x_2)$ and send $y_i$ to $P_i$.

Figure 5: Ideal functionality of secure two-party computation.

computation constructions [46, 16]. In such settings, the circuit's structure (and hence the algorithm it encodes) is public, even if the inputs remain hidden.

## 5 Experimental Evaluation

We implement https://github.com/Gefei-Tan/SURE and evaluate the efficiency of *SURE* in repairing Stable Diffusion v1.4 [34] with 32 layers, and compare it with a baseline model repair protocol that runs entirely within a generic 2PC framework for comparison.

**Experiment Setup**. We implement our end-to-end protocol *SURE* and compare it with a baseline protocol that executes all editing operations within a generic 2PC framework. The baseline is implemented based on the `semi` protocol variant from the `MP-SPDZ` framework (BSD3 License) [22], a popular framework for benchmarking generic secure protocols. However, `MP-SPDZ` does not provide a low-level OT interface suitable for our customized protocol in *SURE*. For ease of integration, we instead implemented *SURE* using the `EMP-OT` library from the `EMP-toolkit` (MIT License) [42], which provides efficient implementations of various OT primitives and a flexible low-level API. To evaluate both *SURE* and the baseline, we perform a single model repair on Stable Diffusion v1.4 [34]. In this model, the source and target prompt embeddings $\mathbf{C}, \mathbf{C}^*$ are matrices of shape $[768, 77]$. The repair algorithm modifies 32 cross-attention layers in total, where each layer contains key and value projection matrices of shape $[320, 768]$. As a result, each fix matrix $\mathbf{W}_{\mathsf{fix}}$ has dimension $[768, 768]$. We set the query and database keys $\mathbf{k}_{\mathsf{qry}}, \{\mathbf{k}_i\}_{i \in [n]}$ to 100-dimensional vectors. We represent all values using single-precision floating-point numbers and use Euclidean distance as the similarity metric for key matching. All experiments are run with a single thread on two Amazon EC-2 `c7i.2xlarge` instances, each with 16 GB of RAM.

**Baseline**. To highlight the efficiency of our lightweight protocol *SURE*, we implement a baseline model repair protocol that runs entirely within a generic 2PC framework for comparison. To ensure a fair comparison, we apply several optimizations to avoid penalizing the baseline unnecessarily. First, we represent editing computation as an arithmetic circuit, which is more efficient than Boolean circuits for linear algebra. Additionally, we use 32-bit fixed-point representation to avoid the high cost of floating-point arithmetic in 2PC. To reduce overhead further, we allow the model repair institute $\mathcal{I}$ to pre-compute the inverse matrices $\mathbf{W}_{\mathsf{inv}}^p = \left( \lambda_p \mathbf{I}_c + \mathbf{C}_p \mathbf{C}_p^\top \right)^{-1}$ outside the 2PC to avoid costly secure matrix inversion. The baseline protocol proceeds as follows:

1. $\mathcal{S}$ inputs the query key $\mathbf{k}_{\mathsf{qry}}$ and all projection matrices $\{\mathbf{W}_K^i, \mathbf{W}_V^i\}_{i \in [m]}$ in all $m$ layers; $\mathcal{I}$ inputs the repair database $\{\mathbf{k}_i, \lambda_i, \mathbf{C}_i, \mathbf{C}_i^*, \mathbf{W}_{\mathsf{inv}}^i\}_{i \in [n]}$.

2. The circuit matches the closest index $p = \arg\min_{i \in [n]} d(\mathbf{k}_{\mathsf{qry}}, \mathbf{k}_i)$ and breaks ties by choosing the smallest $i$. Then, for each layer $i \in [m]$, it computes the fix for all matrices:

$$\mathbf{W}_\star^{\prime i} \leftarrow \left( \lambda_p \mathbf{W}_\star^i + \mathbf{W}_\star^i \mathbf{C}_p^* \mathbf{C}_p^\top \right) \mathbf{W}_{\mathsf{inv}}^p \quad \star \in \{K, V\}.$$

3. The updated matrices $\{\mathbf{W}_K^{\prime i}, \mathbf{W}_V^{\prime i}\}_{i \in [m]}$ are then revealed to $\mathcal{S}$.

Despite these optimizations, the baseline remains orders of magnitude slower than *SURE*.

**Efficiency**. Figure 6 shows the end-to-end runtime and communication cost of *SURE* when performing a full Stable Diffusion v1.4 repair across varying repair database sizes. *SURE* completes the repair in under 17 seconds, even with a repair database of 3,000 entries, with communication capped at 17.1 GB. Therefore, *SURE* **is highly efficient**.

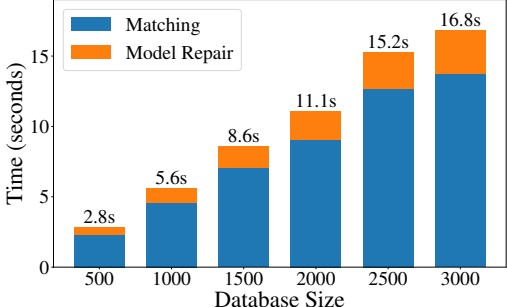 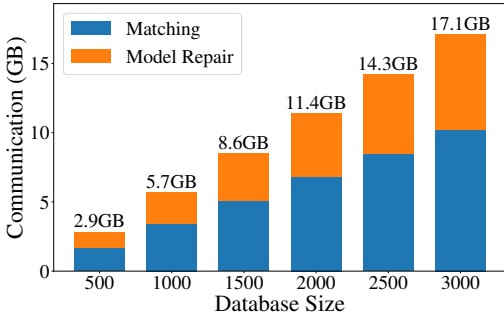

Figure 6: **Runtime and communication cost for *SURE* to repair Stable Diffusion v1.4 with varying repair database sizes.** Reported times are averages over 10 runs. Each database entry consists of a [768,768] fix matrix and a [100,1] key, where all numbers are single-precision floating-point. Communication reported is the larger of the two parties' data sent. Costs are decomposed into two stages: (1) **Matching** – finding the closest key in the database to the failure query using the Euclidean distance as the similarity metric and (2) **Model Repair** – returning the fix via the oblivious transfer protocol to $\mathcal{S}$.

Table 1: **Runtime and communication costs of *SURE* vs. the baseline approach to perform one repair of Stable Diffusion v1.4 with varying database sizes.** *SURE* is orders of magnitude faster than the baseline, as our protocol completely avoids matrix operations in secure computation.

| Database Size | Baseline | | Ours | | |
| --- | --- | --- | --- | --- | --- |
| | **Running Time (hours)** | **Comm. (TB)** | **Running Time (seconds)** | **Comm. (GB)** | **Running Time Improvement** |
| 500 | 167.36 | 76.42 | 2.81 | 2.94 | $\mathbf{2.14 \times 10^5}$ |
| 1000 | 171.84 | 82.30 | 5.58 | 5.65 | $\mathbf{1.11 \times 10^5}$ |
| 1500 | 176.02 | 88.19 | 8.62 | 8.61 | $\mathbf{7.35 \times 10^4}$ |
| 2000 | 180.48 | 94.08 | 11.10 | 11.41 | $\mathbf{5.85 \times 10^4}$ |
| 2500 | 184.96 | 99.97 | 15.21 | 14.32 | $\mathbf{4.38 \times 10^4}$ |
| 3000 | 189.44 | 105.86 | 16.77 | 17.09 | $\mathbf{4.07 \times 10^4}$ |

**Benchmarking.** We further break down the total cost into two main stages–key matching and model repair–as described in our protocol in Figure 3. **Most of the runtime is spent in the matching stage**, which uses a lightweight 2PC protocol to identify the nearest key. In contrast, the OT-based model repair phase is highly efficient and remains nearly constant regardless of database sizes.

**Scalability.** Our protocol scales well with both the repair database and model size: since only a single fix matrix is retrieved and applied across all layers, **the online runtime is independent of the number of model layers**. Moreover, *SURE*'s modular design allows for further optimization: the matching step can be replaced with more efficient cryptographic primitives such as private information retrieval or fuzzy private set intersection. In cases where the database key is public[5], the matching phase can be skipped entirely to further reduce overhead.

**Comparison.** Table 1 compares our protocol against the baseline. **Our protocol achieves up to a $2 \times 10^5$ speedup**. This dramatic improvement stems from avoiding expensive matrix operations and linear scans within 2PC. In the baseline, most of the cost arises from executing the entire editing formula securely and retrieving the correct fix matrix through a full scan of the database, both of which scale poorly with the database size. In contrast, our customized design isolates the secure computation to a small matching task and a lightweight OT-based retrieval protocol, while offloading all matrix operations to local (offline) computation, resulting in far superior performance.

# Acknowledgments

Work of Gefei Tan and Xiao Wang is partially supported by NSF awards #2318975 and #2236819.

---

[5]For example, [30] considers a database of gender bias in different professions, where the database key is simply the profession name (e.g., "nurse"), which is made public. Because the index of the desired fix is known in advance, matching is unnecessary.

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

# Appendices

## A   Limitations

*SURE* is the first secure framework for model editing and demonstrates high efficiency even on commercial-scale text-to-image diffusion models. However, it is currently limited to this domain. Extending to other models, such as LLMs, may require new editing algorithms and cryptographic techniques. Also, our efficient protocol only supports linear editing of the model weight; if editing techniques involve architectural changes or non-linear weight updates, further optimizations might need to be made to maintain its efficiency. While *SURE* can handle multiple repairs, no optimization, batching, or amortization is implemented. We comment that the matching phase can be further optimized for multiple repair settings using techniques like fuzzy private set intersection or private information retrieval.

## B   Ideal Functionalities

The ideal functionality of 1-out-of-$n$ OT is depicted in Figure 4. This can be efficiently realized using $\log n$ 1-out-of-2 OT, which can in turn be efficiently computed using existing cryptographic protocols. The ideal functionality of 2PC is presented in Figure 5.

## C   On the Utility and Scope of Supported Editing Algorithms of *SURE*

In this section, we clarify the relationship between our cryptographic protocol *SURE* and the underlying model editing algorithms it is designed to protect. We detail the preservation of editing utility and discuss the class of algorithms our framework supports.

### C.1   *SURE* Preserves the Utility of the Underlying Editing Algorithm

A central point of our work is that *SURE* is a cryptographic protocol, not a new model editing algorithm. Its purpose is to execute an existing editing algorithm in a secure, privacy-preserving manner using two-party computation (2PC). Our protocol does not alter the underlying mathematical operations of the editing algorithm itself. Additionally, in our implementation, all numerical values are represented using standard single-precision floating points, which ensures that the numerical accuracy is identical to the original, non-private computation. Consequently, the utility, efficacy, and potential side effects of an edit performed using *SURE* are identical to those of the original algorithm (e.g., TIME [30] or UCE [14]). All quantitative and qualitative evaluations from the original papers—such as editing effectiveness, generalization, concept specificity, and impact on general image quality (FID/CLIP scores)—are directly applicable to edits performed with our method.

### C.2   *SURE* Supports More Advanced Algorithms

Our protocol is designed to efficiently support a general class of editing algorithms: any algorithm that can be expressed as a linear transformation of the model's weights. As formalized in the main body of our paper, this class includes any editing algorithm where the update to a layer's weight matrix $\mathbf{W}$ can be refactored into a matrix multiplication with a "fix matrix" $\mathbf{W}_{\text{fix}}$.

Crucially, our framework supports the more powerful UCE algorithm [14] without incurring any additional computational overhead. UCE is capable of performing complex batch edits, such as debiasing multiple attributes or erasing up to 100 artistic styles simultaneously. Its closed-form update rule (equation 7 in the paper) is given as:

$$\mathbf{W}' = \Big( \sum_{c_i \in E} v_i^* c_i^\top + \sum_{c_j \in P} \mathbf{W}_{\text{old}} c_j c_j^\top \Big) \Big( \sum_{c_i \in E} c_i c_i^\top + \sum_{c_j \in P} c_j c_j^\top \Big)^{-1}$$

where $E$ and $P$ are sets of editing target and preserving target and $v^* = \mathbf{W}_{\text{old}} c_i^*$. We can refactor the formula in a similar way by plugging in $v^*$:

$$\mathbf{W}' = \mathbf{W}_{\text{old}} \underbrace{\Big( \sum_{c_i \in E} c_i^* c_i^\top + \sum_{c_j \in P} c_j c_j^\top \Big) \Big( \sum_{c_i \in E} c_i c_i^\top + \sum_{c_j \in P} c_j c_j^\top \Big)^{-1}}_{\mathbf{W}_{\text{fix}}},$$

where $\mathbf{W}_{\mathsf{fix}}$ is the terms in brackets. Notice that, like TIME, computing the $\mathbf{W}_{\mathsf{fix}}$ is independent of $\mathbf{W}_{\mathsf{old}}$, and therefore can be prepared by the repair institute before any interaction.

While TIME and UCE apply a single, global fix matrix to all edited layers, our protocol is not restricted to this paradigm. Our framework can easily and naturally generalize to support layer-specific fix matrices with only negligible additional overhead. In this scenario, the dominating key matching phase of our protocol remains constant, while only the Oblivious Transfer (OT) cost increases minimally with the number of distinct matrices.

## D  Extending *SURE* to Malicious Security

We comment that our protocol's modular design allows for a direct extension to the malicious security setting. The overall protocol structure will largely remain the same and consists of two minor changes: (1) replacing the semi-honest cryptographic primitives with their maliciously secure counterparts and (2) adding a lightweight consistency check to ensure the potentially malicious Service Provider will provide the same index it retrieved from the matching phase to the OT functionality. Because the matching circuit and number of oblivious transfers are both very small, switching to their malicious version will not blow up the overall runtime. Additionally, the consistency check has a constant cost independent of the database size.

We estimate the running time of malicious secure *SURE* using malicious OT and 2PC subprotocol, and implement the malicious version of the baseline using the maliciously secure protocol variant `mascot` from the MP-SPDZ. Table 2 shows the runtime breakdown of both our semi-honest and malicious protocols compared to the baseline implementation. Maliciously secure *SURE* increases the total runtime by roughly $9\times$ compared to the semi-honest one. The overhead is almost entirely incurred by the Matching phase, which runs in a malicious 2PC. In contrast, the Model Repair phase, which only requires a few oblivious transfers, incurs very little additional cost. Malicious security overhead can be avoided in scenarios where the matching phase is not needed. In such cases, the protocol only needs a few malicious OTs and can skip the expensive 2PC and the consistency check; the cost of malicious security introduced by $\mathcal{O}(\log n)$ malicious OT becomes negligible.

Table 2: **Runtime of *SURE* with semi-honest and malicious security vs. the baseline approach to perform one repair of Stable Diffusion v1.4 with varying database sizes.** Maliciously secure *SURE* increases the total runtime by roughly $9\times$ compared to the semi-honest version.

| Database Size | Baseline | Ours | | | | Improvement |
|---|---|---|---|---|---|---|
| | Malicious | Malicious | | Semi-honest | | |
| | Total (h) | Repair (s) | Total (s) | Repair (s) | Total (s) | |
| 500 | 2860.91 | 0.55 | 23.35 | 0.54 | 2.82 | $4.41 \times 10^5$ |
| 1000 | 2937.49 | 1.08 | 46.61 | 1.05 | 5.60 | $2.27 \times 10^5$ |
| 1500 | 3008.95 | 1.58 | 72.19 | 1.53 | 8.60 | $1.50 \times 10^5$ |
| 2000 | 3085.95 | 2.11 | 92.53 | 2.05 | 11.09 | $1.20 \times 10^5$ |
| 2500 | 3161.77 | 2.66 | 129.19 | 2.59 | 15.24 | $8.81 \times 10^4$ |
| 3000 | 3238.35 | 3.17 | 140.53 | 3.09 | 16.83 | $8.30 \times 10^4$ |

