# OpenReview forum: "Pin the Tail on the Model: Blindfolded Repair of User-Flagged Failures in Text-to-Image Services"
_NeurIPS.cc/2025/Conference — NeurIPS 2025 poster_

### Official Review · Reviewer_xN9B · 2025-06-27

**Clarity:** 4
**Significance:** 4
**Originality:** 4
**Rating:** 4
**Confidence:** 4

**Summary:**

This paper proposes a novel privacy-preserving knowledge-sharing framework for knowledge editing in text-to-image models, addressing the scenario involving a model developer and a repair institute. In this context, the institute aims to conceal proprietary repair techniques, while the developer seeks to avoid disclosing model weights. To address these privacy constraints, the authors introduce a privacy-preserving model repair methodology that leverages Oblivious Transfer (OT) as a core mechanism to safeguard the confidentiality of both parties. However, the direct application of OT suffers prohibitively high computational overhead due to its inherent complexity. To mitigate this limitation, the framework optimizes the process by decomposing existing model repair methodologies (TIME) into modular components, thereby significantly reducing computational costs while preserving the privacy guarantees of OT.

**Questions:**

1. could you further elaborate why service providers can not be the a model repair institute? In other word, what is the main challenge of the model repair task?

2. Can this method be applied to erasing method that does not have a closed-form solution? Can this method be applied to methods with very complex closed-form solution?

3. What is the effectiveness of the proposed method? In detail, is the encrypted editing method (TIME) still works? More qualitative and quantitative results should be provided.
______________
Overall, I find the proposed scenario practical and interesting. However, I'm still not sure whether it is a **real problem** in the world (question 1). Besides, I'm not sure if more complex editing methods can be encrypted (question 2 and 3). I will raise my score if these concerns are addressed.

**Ethical Concerns:**

["NO or VERY MINOR ethics concerns only"]

**Final Justification:**

I lean towards acceptance as this paper proposes an interesting while very practical concerns in the real world. However, I’m hesitant give a higher score as the problem formulation is not very rigorous and the provided solution is somewhat narrow.

**Limitations:**

The authors have already stated the limitations of their proposed method.

**Quality:**

3

**Strengths And Weaknesses:**

## Strengths

+ The proposed framework demonstrates significant novelty and practical orientation, addressing a pressing issue that closely mirrors real-world challenges in model deployment.

+ The method to disentangle the update weight in concept erasing method (TIME) is clever and it can help to improve the efficiency of the privacy preserving method.

## Weaknesses

+ I find the paper not very friendly to those who are not familiar with cryptography. I can follow each step of the proposed method. However, I still do not understand how privacy is preserved. I suggest adding more preliminaries about this to facilitate those who are not familiar with cryptography.

+ The proposed method’s reliance on closed-form solutions for model editing restricts its applicability. LoRA-based methods or approaches requiring iterative weight updates (e.g., fine-tuning) cannot be directly supported. Even for techniques with closed-form solutions, decomposition into lightweight and precomputed components may fail if the closed-form equations are inherently complex. This narrow scope limits the framework’s potential for broader adoption in diverse editing scenarios.

+ Though the scenario described in the intro is very practical, I still find several defaults. First, why service providers can not repair their models? (I'm not convinced by the reasons given in line 35-38). Model repairing (or editing, erasing) is not a complex task (at least in this paper, TIME is not a complex method). Given this, why service providers can not repair their models?

+ The assumption that model service provider lacks the ability to perform repairs requires stronger justification. Given that concept-erasure methods like TIME are quite straightforward (e.g., matrix operations), the assumption that model service provider would outsource such tasks to third parties seems under-supported. The authors should provide empirical evidence (e.g., case studies) or domain-specific constraints to validate this scenario.

+ typos: line 2 test $\rightarrow$ text

---

> ### Author Rebuttal · Authors · 2025-07-30
>
> We thank you for noting the strengths of our paper, namely:
> - Significant novelty and practical orientations
> - Addressing a pressing problem with real-world challenges
> - Clever optimizations
>
> We fixed all typos and respond below to all questions raised.
>
> ### **paper is not very friendly to those who are not familiar with cryptography**
> Thank you for your suggestions! We will add more crypto preliminaries in the text. Here's an intuitive summary of how privacy is preserved and the related concepts.
>
> **Secure Multi-Party Computation (MPC):** MPC enables multiple parties to jointly compute an agreed-upon function of their inputs. The protocol guarantees that no information beyond the output is revealed. 2PC (Two-Party Computation) is the specialized case for two participants.
> We use a 2PC protocol for the Matching Phase. The 2PC matching function takes the provider's matching key (semantically, “fix gender bias in nurse”) and the institute's set of database keys as inputs. It only reveals the index of the correct match to the service provider. The index is later used to retrieve the fix matrix via OT.
>
> **Oblivious Transfer (OT):** Oblivious Transfer is a protocol between a sender and a receiver. The sender has multiple messages, and the receiver wants to retrieve one of them. OT’s privacy guarantee ensures that (1) the sender learns nothing about which item the receiver chose, and (2) the receiver learns nothing about the items they didn't choose.
> Our protocol uses OT in the Model Repair phase. The service provider (receiver) privately retrieves the specific $W_{fix}$ matrix it needs from the repair institute's (sender's) database without revealing which one it picked.
>
> **Semi-Honest and Malicious Security:** A semi-honest (or honest-but-curious) adversary correctly follows the protocol specification but may attempt to infer additional information. A malicious adversary may arbitrarily deviate from the protocol. We implement and evaluate our protocol with semi-honest security, but our protocol can also be extended to malicious security without significant overhead. We estimated the maliciously secure version of our protocol and present the numbers in the table below.
>  Database Size | **Repair Only (Malicious)** | Repair Only (Semi-Honest) | **Matching+Repair  (Malicious)** | Matching+Repair (Semi-Honest) | **Matching+Repair (Baseline,Malicious)**
> ---|---|---|---|---|---
>  500 | 0.55s | 0.54s | 23.35s | 2.82s | 2860.91h
>  1000 | 1.08s | 1.05s | 46.61s | 5.60s | 2937.49h
>  1500 | 1.58s | 1.53s | 72.19s | 8.60s | 3008.95h
>  2000 | 2.11s | 2.05s | 92.53s | 11.09s | 3085.95h
>  2500 | 2.66s | 2.59s | 129.19s | 15.24s | 3161.77h
>  3000 | 3.17s | 3.09s | 140.53s | 16.83s | 3238.35h
>
> **How privacy is preserved:** Our protocol works in two main steps: (1) private matching via 2PC and (2) private fix retrieval via OT. The privacy is preserved directly through the security properties of 2PC and OT primitives. Our protocol is formally proven secure under the standard simulation paradigm.
> The core idea is to first define a hypothetical, ideal-world trusted party that handles the entire model repair (Figure 2). By definition, this process is perfectly secure. Then, the proof shows that our real-world protocol “behaves the same” as the ideal-world trusted party. Since the ideal functionality is secure by definition, it follows that our protocol is just as secure and leaks no unwanted information.
>
> ### **Why service providers can't be repair institute? What is the challenge of the model repair task?**
> The primary challenge in model repair is not the computational cost of applying a fix but the accurate diagnosis of a failure from limited user feedback. We argue it is challenging for service providers to repair their model for several key reasons:
> 1. **Insufficient Feedback for Diagnosis:** Modern generative models typically rely on simple feedback mechanisms like a "thumbs up/down" button[Ref. 2]. This single bit of information is ambiguous and lacks the context needed to identify a specific failure. Without extensive expertise of user reports from the repair institute, interpreting a "thumbs down" is guesswork.
> 2. **Lack of Specialized Domain Knowledge:** Even with feedback, service providers cannot possess the deep domain expertise required to pinpoint subtle errors. They may not be familiar with the nuances of niche topics (e.g., the inaccurate details of a sports car), the evolving standards for evaluating social bias, or the specifics of a factual inaccuracy. They can't fix a problem they don't have the expertise to understand.
> 3. **Inability to Keep Knowledge Current:** A model's implicit knowledge can become outdated overnight due to news, cultural events, or even a celebrity changing their hairstyle, or other significant events after its knowledge cutoff. It is not the core business of a service provider to track these real-world changes, making it difficult for them to identify and repair obsolete information.
>
> For example, consider a user who prompts for an image of the "latest iPhone" shortly after a new model is released and gives a "thumbs down" to the generated image of an older model. A service provider, seeing a plausible modern smartphone, would be unable to diagnose the error from this vague feedback. They may lack the knowledge to distinguish the subtle design changes between models, and it is not their core business to track real-time product releases. In contrast, a specialised repair institute would immediately recognize that the model's knowledge is outdated and provide the correct fix.
>
> ### **Can this method be applied to erasing method that does not have a closed-form solution? Can this method be applied to methods with very complex closed-form solution?**
> Our method cannot be directly applied to iterative methods, as our focus is on creating a protocol that is both crypto-friendly and highly efficient. While such methods could theoretically use generic secure computation, our baseline shows that even simple edits become prohibitively inefficient, making iterative approaches infeasible in practice.
> However, we note that our framework is highly versatile and supports advanced, state-of-the-art editing methods. Closed-form editing solutions handle erasing tasks very well, and our protocol supports highly complex closed-form solutions. The core principle is that as long as the final weight update can be expressed as $W’ = WW_{fix}$, our protocol can apply it, regardless of the complexity involved in deriving $W_{fix}$. The follow-up work on TIME titled "Unified Concept Editing in Diffusion Models" (UCE) [Ref 1] is the perfect example of this. UCE provides a more complex closed-form solution that is capable of repairing multiple targets in a single edit, which contains a set for desired fixes and a specific “preservation term” to further prevent undesirable editing artifacts. We observed that the closed-form formula of UCE can also be refactored, and the fix can be encapsulated into a fix matrix $W_{fix}$ of the same size. Therefore, our protocol can directly support it without additional overhead.
>
> Specifically, UCE gives the following closed-form update formula (equation 7 in the paper):
> $W =\Bigl(\sum_{c_i\in E} v_i^* c_i^\top+\sum_{c_j\in P} W_{{old}} c_j c_j^\top\Bigr)\Bigl(\sum_{c_i\in E} c_i c_i^\top+\sum_{c_j\in P} c_j c_j^\top\Bigr)^{-1}$
> where $E$ and $P$ are sets of editing target and preserving target and $v^* = W_{old}c_i^* $.
> We can refactor the formula in a similar way by plugging in $v*$
> $W = W_{{old}}\Bigl(\sum_{c_i\in E} c_{i}^* c_i^\top+\sum_{c_j\in P} c_j c_j^\top\Bigr)\Bigl(\sum_{c_i\in E} c_i c_i^\top+\sum_{c_j\in P} c_j c_j^\top\Bigr)^{-1}$ ,
> where $W_{fix}$ is the terms in brackets.
> Notice that, like TIME, computing the $W_{fix}$ is independent of $W_{old}$, and therefore can be prepared by the repair institute before any interaction.
>
> ### **Effectiveness of the proposed method: does the encrypted editing method (TIME) still works? Need qualitative and quantitative results**
> Our method is instantiated by TIME and can also support its more powerful follow-up UCE as discussed above. Crucially, our method does not change the underlying editing algorithm itself. In our experiment, all numbers are represented using standard single-precision floating point, so the numerical accuracies will be identical as well. Therefore, the utility and effectiveness of an edit applied via SURE are identical to those of the original algorithm, and the comprehensive evaluations of the original TIME and UCE paper can be directly adopted to conclude the effectiveness of SURE, which we briefly summarize below.
>
> Qualitatively, the TIME algorithm is shown to work on a wide range of tasks. It can change object attributes like making roses blue, update factual information such as making a character blond, and apply various stylistic changes. Quantitatively, TIME demonstrates high efficacy in making the intended edit and strong generality in applying that change to related concepts. While there is a slight trade-off, it largely maintains specificity, leaving unrelated images untouched. Importantly, standard image quality metrics like FID and CLIP scores remain stable, which means that the overall model quality is preserved. Furthermore, our protocol supports the more advanced UCE algorithm, which extends these capabilities to complex tasks. UCE can perform multi-attribute debiasing for gender and race and is powerful enough to erase up to 100 artistic styles simultaneously without significant degradation to the model.
>
> [Ref 1] Gandikota, R., Orgad, H., Belinkov, Y., Materzyńska, J., & Bau, D. Unified concept editing in diffusion models. In Proceedings of the IEEE/CVF Winter Conference on Applications of Computer Vision, 2024.
>
> [Ref. 2] Collins et al. (2024). Beyond Thumbs Up/Down: Untangling Challenges of Fine-Grained Feedback for Text-to-Image Generation.

---

> > ### Comment · Reviewer_xN9B · 2025-08-04
> >
> > Sorry for this late reply. I have carefully considered the authors’ rebuttal and the comments from the other reviewers. While I appreciate the clarity provided in the response, I maintain my overall assessment of weak accept:
> >
> > I acknowledge that the paper addresses a timely and practically relevant problem and I value the motivation behind the work. However, two key concerns remain unresolved to my satisfaction:
> >
> > - Justification of the threat model and practical assumptions: The explanation regarding why service providers cannot perform model repair themselves is still not fully convincing. While the point about insufficient feedback for diagnosis is reasonable, I remain skeptical of the broader assumption that model repair must be outsourced to a third party. In practice, many service providers are also the model developers (e.g., large AI companies), and thus have full access to training data, model architecture, and intermediate states. The argument that model repair requires specialized expertise or is inherently too complex to be handled in-house needs further clarification. Without clearer justification, the assumed separation between service provider and repair institute appears artificial.
> >
> > - Limited applicability to modern architectures: The proposed method relies on closed-form solutions for unlearning, which currently restricts its applicability to certain model families—specifically, U-Net-based diffusion models (e.g., Stable Diffusion 1.x–XL). However, recent advances have shifted toward transformer-based architectures such as Diffusion Transformers (DiT) used in SD3 and FLUX. These do not admit the same closed-form solutions, which limits the method’s relevance to state-of-the-art models. The authors should more explicitly discuss these limitations and consider potential pathways toward generalization, even if speculative.
> >
> > In summary, while the paper presents a principled approach to a realistic problem, the assumptions and scope need to be better contextualized within current deployment practices and architectural trends. I believe the work is promising but would benefit from a more critical discussion of its practicality and scalability in modern settings.

---

### Official Review · Reviewer_b8Yo · 2025-07-01

**Clarity:** 3
**Significance:** 3
**Originality:** 3
**Rating:** 4
**Confidence:** 3

**Summary:**

This paper introduces SURE, a novel end-to-end framework capable of securely and blindly repairing post-deployment failures, such as inconsistent image generations caused by outdated, incorrect, or biased assumptions. SURE adopts a co-design approach, optimizing machine learning and cryptographic components to facilitate confidential collaboration between service providers and model repair institutes. By refactoring the editing algorithm into a "crypto-friendly" form and utilizing a customized two-party computation (2PC) protocol combined with oblivious transfer (OT), SURE avoids expensive matrix operations in secure computation, offloading a significant amount of computation offline. Experiments demonstrate remarkable practical efficiency, achieving a four-order-of-magnitude improvement over a generic 2PC baseline. This work successfully demonstrates the feasibility of practical and secure model repair for large-scale diffusion services, ensuring continuous alignment with evolving societal expectations without compromising sensitive data or proprietary knowledge.

**Questions:**

1.Questionable Generality of W_fix: How generalizable is the "One fix fits all" claim for W_fix across diverse repair tasks beyond simple bias correction (e.g., object removal, style alteration), and what experimental evidence supports its non-destructive application to such varied failures?
2.Unevaluated Impact on Unrelated Generation Tasks: What is the impact of modifying all 32 cross-attention layers on the model's general generative capabilities for unrelated prompts, and do these global modifications introduce unintended artifacts or reduce quality for unaddressed tasks?
3.Lack of In-depth Qualitative Evaluation and Repair Examples: Could the paper provide detailed qualitative evaluations, including before-and-after image comparisons, to demonstrate the visual quality, semantic consistency, and lack of new artifacts in repaired images, as quantitative metrics alone are insufficient?
4.Insufficient Verification of Failure Type Diversity: How does SURE's effectiveness generalize to a broader range of failure types beyond bias correction, such as outdated information, factual inaccuracies, or stylistic changes, and what experimental evidence supports this versatility?
5.Scalability and Maintenance Costs of the Repair Database: What are the practical implications and costs associated with scaling and maintaining the "repair database," particularly concerning its construction, efficient retrieval, and conflict resolution when multiple W_fix entries apply or conflict?
6."Blindfolded Repair" Context and Limitations: Given the title "Blindfolded Repair," could the paper more rigorously define the scope of privacy and the adversarial models covered, especially concerning malicious attackers or sensitive PII in user feedback?

**Ethical Concerns:**

["NO or VERY MINOR ethics concerns only"]

**Final Justification:**

After reading the rebuttals and other reviewers' comments, I think this is a good paper but there are still some drawbacks to be solved. Thus I kept my score as  Borderline Accept.

**Limitations:**

Yes

**Quality:**

3

**Strengths And Weaknesses:**

strengths：
1. This paper innovatively introduces the SURE framework, offering the first end-to-end solution for addressing "failures" (such as biases or outdated knowledge) that inevitably arise in large-scale diffusion models during real-world deployment. Crucially, it achieves this while preserving the confidentiality of user feedback, the Service Provider's proprietary model, and the Model Repair Institute's specialized knowledge. This is a groundbreaking problem with significant practical implications and academic value in the current landscape of widespread deep learning model application.
2.The SURE framework achieves secure repair of diffusion models through the co-design of a model editing algorithm with customized Two-Party Computation (2PC) protocols. This "co-design" approach is a core technical contribution of the paper, enabling extremely high repair efficiency while maintaining privacy. Experimental results demonstrate that SURE achieves orders-of-magnitude speed improvements compared to generic baselines, proving the practicality and efficiency of its lightweight protocol.
3.SURE implements model repair by identifying and modifying specific parameters, namely the key and value projection matrices (W_fix) within the diffusion model's cross-attention layers. This strategy, which targets specific parameters, effectively localizes the repair operation to the critical parts influencing model behavior, demonstrating a deep understanding and sophisticated application of the internal mechanisms of diffusion models.

Weaknesses:

1.Questionable Generality of W_fix: The paper claims "One fix fits all," implying that the same W_fix matrix can be reused across all cross-attention layers. However, this "one-shot" assertion requires more detailed explanation and rigorous validation. The paper primarily uses the specific bias repair of "CEO generating female CEO" as an example but does not sufficiently explore or experimentally demonstrate whether a W_fix generated for one specific type of error (e.g., removing a certain object) can be effectively and non-destructively applied to other vastly different repair tasks (e.g., altering image style or correcting factual inaccuracies). Different types of "failures" may necessitate different kinds of knowledge modifications, and whether a single W_fix structure can handle all cases efficiently and precisely remains an unproven claim. If the generality of W_fix is limited only to repairs within specific semantic categories, then the claim of "one fix fits all" appears overly broad.
2. SURE repairs problems by modifying all 32 cross-attention layers of the diffusion model. Such global modifications could have side effects on the model's image generation capabilities for unrelated or non-repair-related text prompts. The paper fails to adequately assess the potential negative impact of the repair operation on the model's overall generation quality, diversity, and unaffected generation tasks. For instance, after fixing the "CEO gender bias," would the model introduce new artifacts or suffer reduced quality when generating "landscape photos" or "animal pictures"? This is a common challenge for all model editing methods, and the paper should provide a more detailed analysis and experimental results.
3.Lack of In-depth Qualitative Evaluation and Repair Examples. Despite claiming that SURE is "highly practical" and offers significant speed improvements, the paper lacks detailed qualitative evaluation of the repaired images. It does not provide comparative examples of images before and after repair, nor does it discuss in detail the visual quality, semantic consistency, or the introduction of new artifacts in the repaired images. Relying solely on quantitative metrics (such as speed) is insufficient to fully demonstrate the effectiveness of the repair method. Providing actual examples of "female CEO" images generated after fixing the "CEO gender bias," along with comparisons to the original problematic images, would greatly strengthen the argument's persuasiveness.
4. The paper primarily discusses model bias, such as gender bias. However, the scope of "failures" can be very broad, including outdated information, factual inaccuracies, removal or addition of specific styles, etc. The paper does not adequately demonstrate SURE's ability to handle and repair diverse types of "failures." For more complex or nuanced repair tasks, SURE's effectiveness remains unknown.
5.The paper mentions that the Model Repair Institute (I) needs to maintain a "repair database." However, there is no in-depth discussion regarding the practical scale of this database, its construction and maintenance costs, or how to ensure efficient retrieval and repair when the number of database entries becomes vast. As the types of model failures continuously increase, the scalability of the database and its associated computational and storage overhead could become a bottleneck for practical deployment. Furthermore, if multiple potential W_fix entries are applicable to the same query, or if different W_fix entries conflict, how does SURE handle these situations?
6.The title "Blindfolded Repair" implies complete privacy for all parties involved. However, this "blindfolded" aspect may not hold true in certain real-world adversarial scenarios, such as when a malicious attacker actively deviates from the protocol to gain information, or in situations where user feedback itself contains highly sensitive Personally Identifiable Information (PII). The paper should more rigorously define the scope of privacy covered by "blindfolded repair" and the adversarial models it addresses.

---

> ### Author Rebuttal · Authors · 2025-07-30
>
> Thank you for your review and valuable feedback!
> We want to clarify a crucial point that addresses your primary concerns (Questions 1-4). Our work introduces SURE, a novel, privacy-preserving protocol for applying model edits, which does not alter the underlying mathematical editing algorithm. Therefore, the utility and effectiveness of an edit applied via SURE are identical to those of the original algorithm. We will add a discussion to clarify this in our manuscript.
> To address your concern regarding the effectiveness of TIME's “one fix fits all” nature and demonstrate how our flexible protocol can benefit from future algorithmic improvements, we consider the follow-up work Unified Concept Editing (UCE) [Ref. 1]. UCE generalizes TIME's closed-form formula to handle multiple, distinct fixes in a single update.  Our protocol fully supports this multi-target editing without loss of efficiency, as the resulting $W_{fix}$ matrix prepared by the repair institute is the same size. Specifically, UCE gives the following closed-form update formula (equation 7 in the paper):
> $W =\Bigl(\sum_{c_i\in E} v_i^* c_i^\top+\sum_{c_j\in P} W_{{old}} c_j c_j^\top\Bigr)\Bigl(\sum_{c_i\in E} c_i c_i^\top+\sum_{c_j\in P} c_j c_j^\top\Bigr)^{-1}$
> where $E$ and $P$ are sets of editing target and preserving target and $v^* = W_{old}c_i^* $.
> We can refactor the formula in a similar way by plugging in $v*$
> $W = W_{{old}}\Bigl(\sum_{c_i\in E} c_{i}^* c_i^\top+\sum_{c_j\in P} c_j c_j^\top\Bigr)\Bigl(\sum_{c_i\in E} c_i c_i^\top+\sum_{c_j\in P} c_j c_j^\top\Bigr)^{-1}$ ,
> where $W_{fix}$ is the terms in brackets.
> Notice that, like TIME, computing the $W_{fix}$ is independent of $W_{old}$, and therefore can be prepared by the repair institute before any interaction.
>
> Given that our protocol faithfully applies these algorithms, the extensive evaluations from the TIME and UCE papers directly answer your questions (1-4) about the editing method's performance. Both papers provide extensive quantitative and qualitative evaluations for (1) efficacy, which measures how effective the editing method is on the editing prompt, (2) generality, which measures how the editing method generalizes to other related prompts, and (3) specificity, which measures the ability to leave the generation of unrelated prompts unaffected. We summarize these results below.
>
> [Ref 1] Gandikota, R., Orgad, H., Belinkov, Y., Materzyńska, J., & Bau, D. Unified concept editing in diffusion models. In Proceedings of the IEEE/CVF Winter Conference on Applications of Computer Vision, 2024.
> ### **Q1: Generality of one fix fits all.**
> Both TIME and UCE show extensively that this closed-form formula is very effective for a wide range of repair tasks (including object removal and style alteration). For example, it could
> - modify object attributes: (e.g., changing a rose's color, a pizza's shape)
> - completely remove concepts (removing a "garbage truck" from the model's knowledge)
> - apply style alteration (erase artistic styles (removing the influence of a specific artist)
> - handle complex, multi-attribute debiasing.
> - perform combinations of these edits simultaneously without a loss in effectiveness.
>
> ### **Q2: Impact on Unrelated Generation Tasks**
> Modifying all layers does not harm unrelated images. The editing is highly targeted to only a small fraction of the model's parameters (2.2% for Stable Diffusion 1.4). In both TIME and UCE, evaluations confirmed that:
> 1. Modifying specific concepts does not affect others. For example, editing "roses" to be blue does not change the colour of "poppies".
> 2. TIME does not degrade the overall generative quality of the model after editing (wrt FID and CLIP Score)
> 3. UCE achieves high fidelity and minimal distortion to generations for unaddressed tasks, even after extensive batch edits.
> 4. UCE’s editing formula contains a specific “preservation term” to further prevent undesirable editing artifacts.
>
> ### **Q3: Lack of In-depth Qualitative Evaluation/Repair Examples**
> The TIME and UCE paper presents extensive before-and-after image comparisons demonstrating its effectiveness, generalizability, which includes changing object attributes ("a pack of roses" to "blue roses"), context ("a cow" to "a cow on the beach"), and correcting biases. They also demonstrate that edits generalize logically to related prompts (e.g., editing "house" also affects "an oil painting of a house"). They also provide visual examples that confirm that unrelated prompts remain pristine (e.g., "poppies" are unchanged when "roses" are edited). We are happy to add them to our paper if you think it is useful.
>
> ### **Q4: Insufficient Verification of Failure Type Diversity**
> The effectiveness and generalisation abilities of TIME and UCE directly apply to SURE. Their evaluations and visual examples show that the editing generalises very well beyond bias correction while having little undesirable artifacts on unaffected prompts.
> 1. Outdated information: TIME fixes outdated information, such as when a celebrity changed their hairstyle. They edit the Harry Potter Franchise character "Hagrid" to become "Blond Hagrid". It shows that it generalizes well for prompts containing the concept of Hagrid (e.g. “a painting of Hagrid”) and doesn't affect other Harry Potter characters.
> 2. Factual Inaccuracies: Both TIME and UCE can override models’ inaccurate "implicit assumptions" about object attributes and align them with desired factual alterations, including altering colors, shape, materials, and completely erasing the concept of an object.
> 3. Stylistic Changes: TIME can induce stylistic shifts by modifying conceptual attributes (e.g., changing the material of a house). UCE can remove specific artistic styles, and experiment shows it can erase up to 100 artists simultaneously before significantly impacting image fidelity or CLIP scores on unrelated concepts.
>
> ### **Q5: costs with scaling repair database: construction, efficient retrieval, and when multiple $W_{fix}$ entries apply or conflict?**
> Regarding scalability, we note that the offline costs for constructing the database are generally negligible compared to the cryptographic online cost, as they only involve standard plaintext matrix operations, which are inexpensive and highly parallelizable on modern hardware.
>
> As for the online cost, the database size will not affect the efficiency of retrieval much (model repair phase in Figure 4). The retrieval only requires $\mathcal{O}(\log n)$ number of oblivious transfers: concretely, for 1 billion entries, it will only require 30 oblivious transfers, although the final running time will depend on the bandwidth, as communication unavoidably will scale linearly with the database size. Database size will mainly affect the cost of the matching phase, but our protocol is modular to handle this: for practical, domain-specific databases of moderate size, a simple linear scan is concretely efficient. For massive-scale databases, the matching component can be swapped with advanced sub-protocols like Private Information Retrieval to achieve sub-linear performance that can easily handle millions of records.
>
> For conflict resolution, we break ties by entry index, mainly for technical consistency. In practice, for a well-curated database with a precise similarity metric, it is unlikely that equally similar entries have significant conflicts in fixes. Furthermore, if a conflicting or low-quality fix were applied, the service provider could readily detect the resulting model degradation through local benchmarking.
>
> ### **Q6: "Blindfolded Repair" Context and Limitations: the scope of privacy and adversarial models, especially concerning malicious attackers or sensitive PII in user feedback.**
> The title "Blindfolded Repair" refers to the privacy guarantee of our protocol: the repair institute provides fixes without learning the model weights or which specific fix a service provider is requesting, and the service provider obtains a needed fix without learning anything else about the repair database or the underlying repairing algorithm.
>
> Our implementation is under the semi-honest threat model (line 122) and we specified our assumption for each party in Section 3. Regarding sensitive PII, we assume the user voluntarily hands their feedback to the service provider, but the service provider must not disclose this to any third party (in practice, this is reasonable under privacy regulations).
> For security against malicious adversaries, our protocol can be easily extended to a malicious setting by (1) replacing cryptographic primitives (2PC and OT) with their malicious secure counterparts and (2) a lightweight consistency check between the matching output and the OT input to ensure a  Service Provider correctly uses the index from the Matching phase in the subsequent steps. We estimate the cost of a maliciously secure version of SURE and compare it with a maliciously secure baseline (implemented using the MASCOT protocol from MP-SPDZ). We summarize the running time in the table below.
>
>
>  Database Size | **Repair Only (Malicious)** | Repair Only (Semi-Honest) | **Matching+Repair  (Malicious)** | Matching+Repair (Semi-Honest) | **Matching+Repair (Baseline,Malicious)**
> ---|---|---|---|---|---
>  500 | 0.55s | 0.54s | 23.35s | 2.82s | 2860.91h
>  1000 | 1.08s | 1.05s | 46.61s | 5.60s | 2937.49h
>  1500 | 1.58s | 1.53s | 72.19s | 8.60s | 3008.95h
>  2000 | 2.11s | 2.05s | 92.53s | 11.09s | 3085.95h
>  2500 | 2.66s | 2.59s | 129.19s | 15.24s | 3161.77h
>  3000 | 3.17s | 3.09s | 140.53s | 16.83s | 3238.35h
>
> Malicious security increases the total runtime by ~9x. This overhead comes almost entirely from the Matching phase executed in malicious 2PC. However, in applications where the Matching phase is not required (as noted in footnote 3), our protocol bypasses the costly 2PC. In this case, the consistency check can also be avoided, and the $\mathcal{O}(\log n)$ numbers of malicious OT adds virtually no overhead.

---

### Official Review · Reviewer_QmFt · 2025-07-03

**Clarity:** 3
**Significance:** 3
**Originality:** 3
**Rating:** 4
**Confidence:** 2

**Summary:**

The paper proposes SURE, which is a light-weight two-party protocol (2PC) that lets a model owner obtain a weight-patch matrix to fix a user-flagged error in diffusion model while keeping weights, user feedback, and the editing method hidden. In this protocol, the repair institute precomputes a fix matrix which the service provider can multiply it to all cross-attention layers locally, which enables a full model repair with small traffic and the protocol is proven secure in the semi-honest model.

**Questions:**

- Have you run any quantitative or qualitative studies to show that a patched model actually corrects the user-flagged error without degrading other outputs?
- How would the protocol/runtime/cost have to change to change to a maliciously secure setting?
- Is there a possibility of a stronger choice of baseline other than a full linear-scan MPC?
- Is it possible to apply several fixes to one prompt at a time? Does the cost increase in this case?

**Ethical Concerns:**

["NO or VERY MINOR ethics concerns only"]

**Final Justification:**

The paper provides a solid method and justification but is limited to the old diffusion model architectures, which somewhat limit my excitements about this work, points being similar to the one raised by reviewer xN9B. Overall, I believe this work provides a solid solution to a niche problem, but is limited in some aspects, therefore I keep my score of weak accept.

**Limitations:**

yes

**Quality:**

3

**Strengths And Weaknesses:**

**Strength**
- The paper tackles a well-motivated problem that is relevant to real-world service providers of diffusion models, as it aims to fix errors in systems that are already deployed while complying to security constraints.
- The paper provides tight security guarantees, with proofs in the hybrid model as well as their algorithm-hiding property of the protocol.
- The method results in a substantially more efficient end-to-end repair efficiency compared to a baseline method with ~17s running time for a 3,000 sized database.

**Weakness**
- The paper does not discuss malicious-security guarantees or expected change of cost (compared to semi-honest setup) but this kind of setting might be required for a real-world use case.
- The paper discusses only Stable Diffusion v1.4, which has a significant gap between more recent generation models that are typically deployed and might need repairs in a real-world setup. Runtime analysis for larger models such as SDXL or SD3 might be necessary to demonstrate generalizability.
- The baseline might be too weak, which might make the comparison a bit unrealistic.

---

> ### Author Rebuttal · Authors · 2025-07-30
>
> Thank you for noting the strengths of our paper:
> - well-motivated and real-world problem;
> - end-to-end efficient solutions with tight security guarantees.
>
> We address your questions below.
>
> ### **The paper does not discuss malicious-security guarantees or expected change of cost (compared to semi-honest setup) but this kind of setting might be required for a real-world use case. How would the protocol/runtime/cost have to change to change to a maliciously secure setting?**
> Our protocol’s modular design allows for a direct extension to the malicious security setting. The overall protocol structure will largely remain the same and consists of two minor changes: (1) replacing the semi-honest cryptographic primitives with their maliciously secure counterparts and (2) adding a lightweight consistency check to ensure the potentially malicious Service Provider will provide the same index it retrieved from the matching phase to the OT functionality.  Because the matching circuit and number of oblivious transfers are both very small, switching to their malicious version will not blow up the overall runtime. Additionally, the consistency check has a constant cost independent of the database size.
>
> We estimate the running time of malicious secure SURE using Malicious OT and 2PC subprotocol. We also report the malicious version of the baseline (using the mascot protocol variant from the MP-SPDZ). Maliciously secure SURE increases the total runtime by ~9x compared to our semi-honest implementation. Note that the overhead is almost entirely incurred by the Matching phase, which runs in a malicious 2PC. In contrast, the Model Repair phase, which only requires a few oblivious transfers, incurs very little additional cost. We comment that malicious security overhead can be avoided in scenarios where the Matching phase is not needed (as we discussed at Line 329, footnote 3). In such cases, the protocol only needs a few malicious OTs and can skip the expensive 2PC and the consistency check; the cost of malicious security introduced by O(logn) malicious OT becomes negligible.
>
>  Database Size | **Repair Only (Malicious)** | Repair Only (Semi-Honest) | **Matching+Repair  (Malicious)** | Matching+Repair (Semi-Honest) | **Matching+Repair (Baseline,Malicious)**
> ---|---|---|---|---|---
>  500 | 0.55s | 0.54s | 23.35s | 2.82s | 2860.91h
>  1000 | 1.08s | 1.05s | 46.61s | 5.60s | 2937.49h
>  1500 | 1.58s | 1.53s | 72.19s | 8.60s | 3008.95h
>  2000 | 2.11s | 2.05s | 92.53s | 11.09s | 3085.95h
>  2500 | 2.66s | 2.59s | 129.19s | 15.24s | 3161.77h
>  3000 | 3.17s | 3.09s | 140.53s | 16.83s | 3238.35h
>
>
> ### **Stable Diffusion v1.4 has a significant gap between more recent generation models that are typically deployed and might need repairs in a real-world setup. Runtime analysis for larger models such as SDXL or SD3 might be necessary to demonstrate generalizability.**
> We note that our work, SURE, is the first privacy-preserving protocol to support complex, real-world diffusion models. We view identifying repair algorithms that can be made crypto-friendly as one of our main contributions, as it's rare for cryptographic protocols to scale to non-trivial, real-world machine learning models of this size without resorting to approximations that would degrade performance. Our protocol achieves this efficiently, performing repairs in **seconds**.
> We chose Stable Diffusion v1.4 specifically because it is a widely-used, large-scale model that serves as a common benchmark in both academic research and deployed products, and it is the benchmark model used in the foundational editing literature (TIME, UCE) that our protocol implements. This allows us to directly validate that SURE applies edits with zero loss of utility compared to the original, non-private algorithms.
> While Stable Diffusion v1.4 serves as our validation target, our protocol is designed to scale efficiently to even larger models. Its scalability can be analyzed by two factors:
>
> **Models with more layers:** A key advantage of our protocol is that its cost is independent of the number of layers being edited. The fix matrix $W_{fix}$ is retrieved once through the interactive protocol, and the service provider can then apply this same fix to all relevant layers of the model locally, without any further interaction. In contrast, the baseline edits each layer with a separate execution of a secure 2PC protocol for matrix operations, which makes the total cost scale linearly.
>
> **Models with larger weight matrices:** Larger models could have higher-dimensional weight matrices, which increase the size of the corresponding $W_{fix}$ . In our protocol, this only impacts the communication cost. The total number of Oblivious Transfer (OT) operations remains the same, but the size of the payload within each OT message increases linearly with the size of the $W_{fix}$.
>
> ### **Have you run any quantitative or qualitative studies to show that a patched model actually corrects the user-flagged error without degrading other outputs?**
>
> The unique feature of TIME is that it does not change the underlying Model editing logic/operation at all. Although we did clear refactoring to the editing algorithm, it only makes it more cryptography-friendly while maintaining the identical effect. Therefore, our method provides the exact same utility as the original algorithm. The quantitative and qualitative studies demonstrating that TIME successfully corrects errors without significant degradation to other outputs can be found in the original paper and its follow-up work.
>
>
> ### **Is there a possibility of a stronger choice of baseline other than a full linear-scan MPC?**
> As we discussed in Appendix C (line 828), our baseline already includes significant optimizations for matrix operations to be a fair point of comparison. Specifically, we allow the repair institute to locally pre-compute parts of the repair formula. This strategy eliminates the most expensive matrix inverse operations and reduces the number of matrix multiplications inside the 2PC protocol. Despite these optimizations, the secure matrix computation remains the overwhelming bottleneck: for a database of 500 entries, the secure edit takes around 167 hours while the linear scan only takes ~500 seconds. This shows that even if we were to use a more customized protocol like Private Information Retrieval to make the matching phase faster, the baseline would remain impractical.
>
>
> ### **Is it possible to apply several fixes to one prompt at a time? Does the cost increase in this case?**
> Thanks for the great question. Yes, our protocol directly supports multiple fixes through a single protocol execution, and the core cryptographic cost does not increase. The only potential cost increase relates to the size of the repair database, which is a manageable trade-off.
>
> Our protocol is compatible with follow-up work on TIME, “Unified Concept Editing in Diffusion Models” [Ref 1], which generalizes the closed-form solution to incorporate multiple editing targets into a single update. Their formula can also be refactored using our optimisations as $W’ = W_{old} W_{fix}$ in a similar way, which makes it compatible with our protocol. Specifically, UCE gives the following closed-form update formula (equation 7 in the paper):
> $W =\Bigl(\sum_{c_i\in E} v_i^* c_i^\top+\sum_{c_j\in P} W_{{old}} c_j c_j^\top\Bigr)\Bigl(\sum_{c_i\in E} c_i c_i^\top+\sum_{c_j\in P} c_j c_j^\top\Bigr)^{-1}$
> where $E$ and $P$ are sets of editing target and preserving target and $v^* = W_{old}c_i^* $.
> We can refactor the formula in a similar way by plugging in $v*$
> $W = W_{{old}}\Bigl(\sum_{c_i\in E} c_{i}^* c_i^\top+\sum_{c_j\in P} c_j c_j^\top\Bigr)\Bigl(\sum_{c_i\in E} c_i c_i^\top+\sum_{c_j\in P} c_j c_j^\top\Bigr)^{-1}$ ,
> where $W_{fix}$ is the terms in brackets.
> Notice that, like TIME, computing the $W_{fix}$ is independent of $W_{old}$, and therefore can be prepared by the repair institute before any interaction.
>
> When applying UCE using our protocol, each $W_{fix}$ encodes multiple editing targets. The cost can be analyzed in two parts:
> The core cryptographic cost of our protocol involves securely retrieving one $W_{fix}$ matrix. Since the multi-edit $W_{fix}$ ​has the same size as a single-edit one, the computation and communication for one run of the protocol do not change at all.
> Increased database size: The potential cost increase comes from the growth of the repair database. To support combined edits, the database must store pre-computed entries for those combinations. For example, a simple database might have entries for fixing gender bias for “nurse”, “engineer”, and “cook”. But to support multiple edits, the database would need to store all possible combinations of these three professions. However, our protocol is modular to make this manageable. For extremely large databases, the default matching component can be swapped with advanced solutions like Private Information Retrieval (PIR), which reduces the matching cost to be sub-linear in the database size.
>
>
> [Ref 1] Gandikota, R., Orgad, H., Belinkov, Y., Materzyńska, J., & Bau, D. Unified concept editing in diffusion models. In Proceedings of the IEEE/CVF Winter Conference on Applications of Computer Vision, 2024.

---

> > ### Comment · Reviewer_QmFt · 2025-08-06
> >
> > I sincerely appreciate the detailed response from the authors. I believe some of my concerns have been addressed, but the fact that more recent larger models have diverged entirely from UNet based models (e.g., SD1.4, SDXL) and towards transformer-based models (e.g., SD3, FLUX) limit my excitement about this work, similarly to the concerns raised by reviewer xN9B. Therefore, I keep my score.

---

### Official Review · Reviewer_zFKo · 2025-07-03

**Clarity:** 3
**Significance:** 3
**Originality:** 3
**Rating:** 4
**Confidence:** 3

**Summary:**

This paper studies the confidential collaboration problem between a diffusion model service provider and a model repair institute. When a diffusion model service user submitted unexpected outputs to the service provider, it's important for the provider to identify and fix such misalignment. While it’s common for a service provider to seek collaboration from external third-party model repair institutes, it’s important to keep the confidentiality of both the provider's proprietary model and user feedback, as well as the repair institutes’ proprietary repair technique. This paper proposes such a secure repairing method, SURE, that encodes the prior Text-to-Image Model Editing (TIME) method into a secure two-party computation (2PC) protocol. The key insight is to identify a decomposition of the computation in TIME to factor out the heavy matrix-multiplication from the 2PC protocol. This paper provides security proof for the proposed method. This paper experimentally evaluated it against a vanilla TIME implementation in 2PC without optimizations, and the results shows that the proposed (optimized) method can significantly reduce the runtime and communication overhead in practice.

**Questions:**

- Regarding the efficiency, what are the assumptions on the editing algorithm to fit this framework to reduce the overhead?

- Regarding the functionality, can this framework support more general repair requests, like align a prompt with a (batch of) images, and general repair methods, like a fine-tuning based method?

- How can another editing method be plugged-in this framework? What are the requirements?

- Can the protocol be extended so that the service provider does not lose access to prior repairs even if the model repair institutes stopped the service? For example, allowing the service provider to store the database and do the database matching offline?

- Minor presentation suggestions:

  - Regarding Table 1, it's recommended to place the caption on the top of a table.

  - If possible, it might be more intuitive to align the unit of time and data for the baseline and SURE in Table 1 to highlight how significant the improvement is. For example, the time unit can be inlined like “167.36h” vs. “2.81s”. This is just a suggestion but not a demand.

**Ethical Concerns:**

["NO or VERY MINOR ethics concerns only"]

**Final Justification:**

The revision plan proposed by the authors has addressed my concerns.

Although being limited to a narrow subset of "crypto-friendly" editing methods, I agree that the proposed idea is novel and solid, and can have practical impact for those applicable architectures and methods. I'll keep my rating as borderline accept.

**Limitations:**

yes

**Paper Formatting Concerns:**

The limitations are not included in the main paper but in the appendix.

**Quality:**

3

**Strengths And Weaknesses:**

## Strengths

- The problem is well-motivated. The problem of efficient and confidential collaboration between service providers and model repair institutes is interesting.

- The idea of incorporating existing editing algorithms with the 2PC protocol to achieve confidential collaboration is novel to this community.

- This work identified an effective optimization in encoding the TIME editing algorithm in a 2PC protocol, which significantly reduced the computation and communication overhead and made the protocol practical.

## Weaknesses

- The proposed framework lacks discussion of assumptions/requirements for a general editing algorithm to fit and gain the overhead reduction. Without such discussion, the proposed method is better to be viewed as an optimized 2PC extension to the TIME algorithm, but not as a general framework.

---

> ### Author Rebuttal · Authors · 2025-07-30
>
> We thank you for noting the strengths of our paper:
> - interesting and well-motivated problems;
> - novel and practical solutions.
>
> We respond below to all questions raised.
> ### **What are the assumptions on the editing algorithm to fit this framework to reduce the overhead?**
> ### **Can this framework support more general repair requests, like align a prompt with a (batch of) images, and general repair methods, like a fine-tuning based method? How can another editing method be plugged-in this framework? What are the requirements?**
>
> **TL;DR: Our protocol assumes the editing can be expressed as a linear transformation of the model’s weight. These editing methods can all be made crypto-friendly through our optimisation. Our protocol directly supports advanced, state-of-the-art editing algorithms like UCE [Ref.1], which can perform batch alignment.**
>
> In theory, it is feasible to execute any editing algorithm securely in 2PC. However, as we show in our baseline, even for TIME, directly plugging it into a 2PC protocol incurs prohibitive costs. To make this practically feasible and mitigate the substantial computational overheads inherent to both 2PC and knowledge editing algorithms, we show that optimisations on both machine learning and cryptography fronts are essential: i) designing a crypto-friendly editing algorithm; ii) developing a customised secure protocol tailored to it.
>
> In this paper, we instantiate one such category of crypto-friendly algorithms using a closed-form repair formula that satisfies two key properties:
> Enabling expensive operations to be performed offline, entirely eliminating matrix operations within the 2PC protocol;
> Being agnostic to the number of layers, allowing the same fix to be reused across all layers.
>
> As formalized in Definition 1, this class includes any editing algorithm that can be refactored as applying a fixed linear transformation to a layer. In particular, we refactor TIME editing algorithm (state-of-the-art in knowledge editing as it is highly efficient and offers a closed-form solution) such that its repair modifies the weight matrix by right-multiplication it with a “fix matrix”: $W’ = WW_{fix}$.
>
> Without incurring any additional costs, our framework can support multiple edits at the same time using "Unified Concept Editing in Diffusion Models" (UCE) [Ref 1]. UCE is capable of repairing multiple targets in a single edit: it can erase up to 100 artists simultaneously before significantly impacting image fidelity or CLIP scores on unrelated concepts. UCE has a more complex closed-form solution that is capable of repairing multiple targets in a single edit, which contains a set for desired fixes and a specific “preservation term” to further prevent undesirable editing artifacts. We observed that the closed-form formula of UCE can also be refactored, and the fix can be encapsulated into a fix matrix $W_{fix}$ of the same size. Therefore, our protocol can directly support it without additional overhead.
>
> Specifically, UCE gives the following closed-form update formula (equation 7 in the paper):
> $W =\Bigl(\sum_{c_i\in E} v_i^* c_i^\top+\sum_{c_j\in P} W_{{old}} c_j c_j^\top\Bigr)\Bigl(\sum_{c_i\in E} c_i c_i^\top+\sum_{c_j\in P} c_j c_j^\top\Bigr)^{-1}$
> where $E$ and $P$ are sets of editing target and preserving target and $v^* = W_{old}c_i^* $.
> We can refactor the formula in a similar way by plugging in $v*$
> $W = W_{{old}}\Bigl(\sum_{c_i\in E} c_{i}^* c_i^\top+\sum_{c_j\in P} c_j c_j^\top\Bigr)\Bigl(\sum_{c_i\in E} c_i c_i^\top+\sum_{c_j\in P} c_j c_j^\top\Bigr)^{-1}$ ,
> where $W_{fix}$ is the terms in brackets.
> Notice that, like TIME, computing the $W_{fix}$ is independent of $W_{old}$, and therefore can be prepared by the repair institute before any interaction.
>
> While both TIME and UCE algorithms use a single fix matrix for all edited layers, our protocol easily and naturally generalises to support layer-specific fix matrices with only negligible additional cost.
> Specifically, the Oblivious Transfer (OT) cost will slightly increase with the number of matrices, but the dominating key matching phase will remain constant. This means the total overhead remains low even when handling multiple distinct fix matrices.
>
> We will summarize and incorporate this clarification into the paper accordingly.
>
> ### **Can the protocol be extended so that the service provider does not lose access to prior repairs even if the model repair institutes stopped the service? For example, allowing the service provider to store the database and do the database matching offline?**
> That is an excellent point regarding the long-term availability of repairs. Our protocol inherently supports this need, although through a different mechanism than the one suggested.
> While the repair database itself is the intellectual property of the repair institute and cannot be shared (see Section 3, line 141), our framework offers a more direct solution. Once the service provider securely obtains the fix through our protocol, they can store it locally and retain it indefinitely. This stored fix can then be reapplied to the current model at any time. Moreover, it can even be applied across multiple different future models that require the same correction, without any further interaction with the repair institute. Therefore, even if the repair institute were to stop its operations, the service provider would not lose access to any previously acquired repairs. This is in contrast to the generic baseline, which requires a costly interactive 2PC execution for every single model repair, even if the fix has been previously applied and the model is the same.
> We appreciate the insight and have highlighted this important property in our problem description.
>
> ### **Minor presentation suggestions**
> Thank you for your careful reading and helpful suggestions regarding the presentation of our paper. We have incorporated both presentation suggestions.
>
>
> [Ref 1] Gandikota, R., Orgad, H., Belinkov, Y., Materzyńska, J., & Bau, D. Unified concept editing in diffusion models. In Proceedings of the IEEE/CVF Winter Conference on Applications of Computer Vision, 2024.

---

> > ### Comment · Reviewer_zFKo · 2025-08-01
> >
> > Thank you for the response.

---

> ### Comment · Reviewer_zFKo · 2025-08-05
>
> I still have concerns about the applicability of the proposed decomposition for most of the model repair methods, which largely limits the proposed decomposition to be a general framework. The proposed decomposition requires the repair method to be expressed as a closed-form parameter update, where most of the computation is in a matrix multiplication that can be decomposed. Such assumption excludes all gradient-descent-based repair methods, and limits the type of repair requests/specification to the ones that are supported by a subset of methods that satisfies this assumption.
>
> Furthermore, the submission is written specifically for TIME, lacks description and discussion as a method-agnostic general framework as well as evaluation on common methods.
>
> Therefore, I suggest the authors to rephrase this work as an efficient 2PC adoption of the prior TIME method instead of a general framework.

---

> > ### Author Response · Authors · 2025-08-06
> >
> > Thank you for the follow-up comments and suggestions. We agree that our framework does not cover all editing methods (e.g., gradient-based ones) and that our use of "general framework" should be more precise. We will revise the manuscript to reflect this.
> >
> > However, we want to clarify that our contribution is more fundamental than a 2PC protocol for a single algorithm.
> >
> > - **Our Contributions:** A central challenge in this field is that most ML algorithms are inherently "crypto-unfriendly," which makes scalable, privacy-preserving protocols for real-world models rare. Therefore, we view our main contribution as two parts: (1) **identifying the specific properties that make an editing algorithm "crypto-friendly"** and (2) **an efficient 2PC framework** optimized for them.  Our work is the first to define this class of algorithms and build an efficient, practical repair protocol for it.
> > - **Applicability of Closed-Form:** We believe this class of editing algorithms is not a niche case, as they can be highly effective. As demonstrated by UCE, closed-form solutions can handle complex, real-world repair tasks. Our framework shows that this entire class can be made practical for private model repair for the first time.
> > - **Broader Impact & Co-Design Target:** By identifying this crypto-friendly property, our work not only supports powerful existing algorithms but also establishes a co-design target for the ML community. Researchers can develop new editing methods that fit this crypto-friendly pattern, knowing a practical and scalable privacy solution already exists.
> >
> > To address your concerns, we will make the following revisions to the paper:
> > 1. **Clarify Our Contributions:** We will make our contribution to focus on a "Framework for Crypto-Friendly Model Editing." We will clarify that one of our main contributions is identifying this class of algorithms.
> > 2. **Clarify the Limitations:** We will move the limitations section into the main body to explicitly discuss which methods we can and cannot support.
> > 3. **Discuss Applicability:** We will add a new section to discuss existing algorithms that can fit into our framework directly and their costs.

---

> > > ### Comment · Reviewer_zFKo · 2025-08-07
> > >
> > > Thank you for the response. The proposed revision plan addresses my concerns.
> > >
> > > Although being limited to a narrow subset of "crypto-friendly" editing methods, I agree that the proposed idea is novel and solid, and can have practical impact for those applicable architectures and methods. I'll keep my positive rating.

---

### Author Response · Authors · 2025-08-01
**Summary of Common Concerns**

We sincerely thank all reviewers for their careful readings, insightful questions, and constructive suggestions. We have provided detailed responses to each reviewer individually and offer this high-level summary of how we have addressed the most common concerns.

### **Editing utility & side effects**
A recurring concern was the evaluation of the repair's effectiveness: whether the repair is successful, does it generalize, and if it introduces undesirable artifacts or degrades overall model quality.

Our central clarification is that **SURE is a cryptographic protocol, not a new editing algorithm.** While our optimization makes SURE exceptionally efficient (repair an entire Stable Diffusion V1.4 model in **seconds**), **it does not alter the underlying mathematical repair algorithm.** Therefore, the utility of an edit performed with SURE is identical to that of the original, non-private algorithm (e.g., TIME and its successor, UCE [Ref 1]).

The extensive quantitative and qualitative evaluations in the original TIME and UCE papers are directly applicable to our method. For example, they demonstrate that edits are highly effective across a diverse range of tasks (object removal, style alteration, bias correction) while preserving general image quality, as confirmed by FID and CLIP scores. We will add a section to our manuscript to make this distinction clear and summarize these key findings.

### **Protocol Flexibility and Supported Algorithms**
Several reviewers asked about our framework's flexibility and ability to address complex failures.
SURE supports any editing method that can be expressed as a linear transformation of the model's weights ( $W’ = W_{old}W_{fix}$ ). **Our protocol fully supports more advanced algorithms like Unified Concept Editing (UCE) [Ref 1], which can perform complex, multi-target repairs.** The UCE editing formula, while more complex, can be refactored into the same $W’ = W_{old}W_{fix}$ form, which makes it compatible with SURE at no additional cryptographic cost. This demonstrates our protocol's ability to leverage the power of state-of-the-art editing techniques for diverse tasks:
- **Complex, Multi-Target Repairs:** Simultaneously removing objects, modifying attributes, and updating factual knowledge in a single operation.
- **Enhanced Specificity and Preservation:** Using an explicit "preservation term" in the formula to actively minimize side effects and ensure edits are precisely targeted.
- **Scalable Style and Concept Erasure:** Erasing up to 100 artistic styles simultaneously without significant degradation to unrelated concepts.

### **Cost to Support Malicious Security**
Several reviewers pointed out the importance of considering malicious adversaries. Our protocol's modular design allows for a straightforward extension to the malicious security setting by (1) substituting the cryptographic modules (2PC and OT) with their malicious secure version and (2) adding a lightweight consistency check. We estimated the performance and result shows **SURE remains practical even in the malicious setting.**
The overhead is almost entirely isolated to the optional matching phase, and our protocol still outperforms a malicious baseline by five orders of magnitude. In scenarios where matching is not needed (e.g., the database index is public, footnote 3), the cost of achieving malicious security becomes negligible. We summarize the results in the table below.
Database Size | **Repair Only (Malicious)** | Repair Only (Semi-Honest) | **Matching+Repair  (Malicious)** | Matching+Repair (Semi-Honest) | **Matching+Repair (Baseline,Malicious)**
---|---|---|---|---|---
 500 | 0.55s | 0.54s | 23.35s | 2.82s | 2860.91h
 1000 | 1.08s | 1.05s | 46.61s | 5.60s | 2937.49h
 1500 | 1.58s | 1.53s | 72.19s | 8.60s | 3008.95h
 2000 | 2.11s | 2.05s | 92.53s | 11.09s | 3085.95h
 2500 | 2.66s | 2.59s | 129.19s | 15.24s | 3161.77h
 3000 | 3.17s | 3.09s | 140.53s | 16.83s | 3238.35h

[Ref 1] Gandikota, R., Orgad, H., Belinkov, Y., Materzyńska, J., & Bau, D. Unified concept editing in diffusion models. In Proceedings of the IEEE/CVF Winter Conference on Applications of Computer Vision, 2024.

---

### Note · Authors · 2025-08-15

We thank all reviewers and ACs for their time and efforts in the reviewing process.

Reviewers highlighted several strengths of our paper, namely:
1. **Novelty:** Our proposed framework is the first end-to-end solution to privacy-preserving model editing; our co-design method of cryptography and ML is noted to have “significant novelty” and "is novel to the ML community." (Reviewer zFKo, b8Yo, xN9B)
2. **Real-world relevance:** Our problem is well-motivated, has practical orientation, and targets realistic deployment settings. (Reviewer zFKo, QmFt)
3. **Efficiency:**  SURE can repair an entire Stable Diffusion model in **17 seconds**, thanks to our optimization which significantly reduces both cryptographic computation and communication overhead. ( Reviewer zFKo, QmFt, b8Yo)
4. **Security:** SURE offers tight, provable security and can be extended to malicious security with modest additional cost. (Reviewer QmFt)

We have answered all individual questions from the reviewers in detail. Below we summarize the main concerns and our clarifications:
1. **What algorithms are supported?** Besides TIME, our framework supports more powerful editing algorithms. In our rebuttal we explained how UCE, a successor of TIME, can be directly plugged into SURE and enable multiple fixes in a single edit without any additional cryptographic overhead.

2. **Is the repair effective?** SURE is a cryptographic framework which does not alter the underlying mathematical repair algorithm. Therefore, the effectiveness of SURE is identical to that of the original, non-private algorithm. Results from TIME and UCE therefore apply directly to SURE: these supported editing algorithms can perform powerful edits across tasks (e.g., object removal, style alteration, bias correction) while preserving overall image quality. In our revision, we will add a short section that (i) makes this separation explicit and (ii) summarizes the relevant TIME/UCE findings.

3. **What is the cost of providing malicious security?** Malicious security ensures privacy even when parties arbitrarily deviate from the protocol. Thanks to our modular design, SURE can be directly extended to support malicious security with small additional overhead: we estimated the cost and showed SURE remains practical: the total runtime increased by ~9x (from 17s to 168s for a 3,000-entry database), and the maliciously secure SURE is still five orders of magnitude faster than a malicious baseline.

---

### Decision · Program_Chairs · 2025-09-17

**Decision:**

Accept (poster)

**Comment:**

The paper received unanimously positive scores from the reviewers. The cite well-motivated framework and problem, proofs, security guarantees, and novelty, to mention a few. There was a healthy discussion with reviewers, which further strengthen the importance of the paper. Hence the decision is to accept. Congrats!